# Control of RAB7 activity and localization through the retromer-TBC1D5 complex enables RAB7-dependent mitophagy

Ana Jimenez-Orgaz[1,†], Arunas Kvainickas[1,†], Heike Nägele[1], Justin Denner[1], Stefan Eimer[1], Jörn Dengjel[2] & Florian Steinberg[1,*]

## Abstract

Retromer is an endosomal multi-protein complex that organizes the endocytic recycling of a vast range of integral membrane proteins. Here, we establish an additional retromer function in controlling the activity and localization of the late endosomal small GTPase RAB7. Surprisingly, we found that RAB7 not only decorates late endosomes or lysosomes, but is also present on the endoplasmic reticulum, *trans*-Golgi network, and mitochondrial membranes, a localization that is maintained by retromer and the retromer-associated RAB7-specific GAP TBC1D5. In the absence of either TBC1D5 or retromer, RAB7 activity state and localization are no longer controlled and hyperactivated RAB7 expands over the entire lysosomal domain. This lysosomal accumulation of hyperactivated RAB7 results in a striking loss of RAB7 mobility and overall depletion of the inactive RAB7 pool on endomembranes. Functionally, we establish that this control of RAB7 activity is not required for the recycling of retromer-dependent cargoes, but instead enables the correct sorting of the autophagy related transmembrane protein ATG9a and autophagosome formation around damaged mitochondria during Parkin-mediated mitophagy.

**Keywords** membrane trafficking; mitophagy; RAB7; retromer; TBC1D5
**Subject Categories** Autophagy & Cell Death; Membrane & Intracellular Transport
**The EMBO Journal (2018) 37: 235–254**

## Introduction

The ancient and evolutionarily conserved retromer complex is an endosomal multi-protein assembly that orchestrates the endocytic recycling of a vast range of transmembrane proteins, either from endosomes to the *trans*-Golgi network (TGN) or from endosomes to the plasma membrane (Burd & Cullen, 2014). The heterotrimeric core of the retromer complex, composed of the proteins VPS26, VPS29, and VPS35 (Seaman *et al*, 1998), localizes to the endosomal membrane via simultaneous binding of the VPS35 subunit to active RAB7-GTP and to the sorting nexin SNX3 (Rojas *et al*, 2008; Seaman *et al*, 2009; Balderhaar *et al*, 2010; Liu *et al*, 2012; Harrison *et al*, 2014). The core retromer also associates with various other proteins of the sorting nexin family to achieve cargo specificity and to deform the endosomal membrane into cargo-enriched tubular carriers (Wassmer *et al*, 2007, 2009; Harterink *et al*, 2011; Temkin *et al*, 2011; Steinberg *et al*, 2013). In addition, the retromer subunit VPS35 also recruits the actin-polymerizing WASH complex to establish an actin-decorated endosomal microdomain from where recycling takes place (Derivery *et al*, 2009; Gomez & Billadeau, 2009). Mutations in retromer components lead to late-onset hereditary Parkinson's disease, which makes a thorough understanding of retromer biology necessary from a medical perspective (Vilarino-Guell *et al*, 2011; Zimprich *et al*, 2011).

Although retromer is known to be a RAB7 effector that depends on this small GTPase for its localization to the endosomal membrane (Seaman *et al*, 2009; Harrison *et al*, 2014), the precise relationship or any potential cross-regulation between retromer and RAB7 has not been investigated in depth. RAB7 is known to be a key organizer/regulator of the late endosomal/lysosomal network that associates with various effectors to orchestrate diverse functions such as lysosomal biogenesis, late endosome–lysosome fusion, autophagosome maturation, and cargo transport within the late endocytic network (Guerra & Bucci, 2016). The activation of RAB7 through the MON1/CCZ1 complex is needed for endosome maturation (Kinchen & Ravichandran, 2010; Nordmann *et al*, 2010; Poteryaev *et al*, 2010; Gerondopoulos *et al*, 2012), whereas several RAB7-specific GAPs have been implicated in autophagy, mitophagy, and the degradative pathway (Frasa *et al*, 2010; Carroll *et al*, 2013; Yamano *et al*, 2014). Whether the activity state of RAB7 or any of its diverse functions are regulated or influenced by the RAB7 effector retromer has remained an unanswered question. Recent work on the RAB7-specific GTPase-activating protein (GAP) TBC1D5 revealed that this GAP forms a tight complex with the

1   Center for Biological Systems Analysis (ZBSA), Faculty of Biology, Albert Ludwigs Universitaet Freiburg, Freiburg, Germany
2   Department of Biology, Fribourg University, Fribourg, Switzerland
   *Corresponding author. Tel: +49 761 203 97198; E-mail: florian.steinberg@zbsa.de
   †These authors contributed equally to this study

retromer subunit VPS29 and that this association is necessary for its GAP activity toward RAB7 (Jia *et al*, 2016). TBC1D5 had previously been proposed to inhibit membrane association of the retromer complex and also regulates trafficking of the retromer cargo GLUT1 during autophagy (Seaman *et al*, 2009; Roy *et al*, 2017). It has also been demonstrated to be involved in the trafficking of the autophagy-related transmembrane protein ATG9a (Popovic & Dikic, 2014). Here, we demonstrate that retromer, together with TBC1D5, serves as a master regulator of RAB7 localization, activity, and mobility, which is needed to maintain deposits of inactive RAB7 on cellular endomembranes. Upon loss of retromer, hyperactivated and immobile RAB7 is sequestered on lysosomes, no longer localizes to damaged mitochondria, and also causes perturbed ATG9a trafficking and defects in mitophagosome formation during Parkin-mediated mitophagy.

## Results

### Retromer controls RAB7 localization

For our work on retromer and RAB7, we utilized a recently released monoclonal antibody targeting human RAB7a to visualize endogenous RAB7a in wild-type HeLa cells and in recently described (Kvainickas *et al*, 2017a) retromer (VPS35 and VPS29)-deficient HeLa cells engineered with CRISPR/Cas9. To our surprise, the localization and distribution of endogenous RAB7a relative to LAMP2-decorated late endosomes/lysosomes was strikingly altered in the VPS35- and VPS29-deficient cell lines. In all three knockout cell lines, RAB7a completely covered the entire LAMP2-decorated late endosomal/lysosomal network, whereas wild-type cells displayed only partial and punctate overlap between RAB7a and LAMP2 (Fig 1A). Interestingly, RAB7a appeared to localize to a large, interconnected network spanning most of the cytosol in wild-type HeLa cells but not in the retromer-deficient cells (Figs 1A and EV1A). Co-staining with mitochondrial (TOM20) and endoplasmic reticulum (PDIA) markers revealed that a substantial portion of endogenous RAB7a localized to mitochondria and the endoplasmic reticulum and also densely covered the TGN46 stained TGN (Fig EV1B–D). Lentivirally expressed GFP-RAB7a also localized to tubular structures which partially overlapped with mitochondrial TOM20, confirming the antibody data (Fig EV1E). To rule out unspecific staining, we also stained HeLa cells that had been transfected with humanized Cas9

and a pool of three RAB7a targeting gRNAs to disrupt the RAB7a gene. This treatment led to a near-complete loss of RAB7 signal in Western blot experiments and also resulted in a complete loss of immunofluorescent signal with the RAB7a antibody, whereas co-stained TOM20 was unaffected (Fig EV1F). To further rule out staining or antibody artifacts, we also mixed wild-type HeLa cells and clonal RAB7a knockout cells 1:1 and stained the mixed population for endogenous Rab7a and LAMP2. Loss of RAB7a led to lysosomal swelling and to a complete loss of the RAB7a antibody signal in the KO cells, whereas RAB7a was robustly detected in wild-type cells immediately adjacent to the KO cells on the same coverslip (Fig 1B). Furthermore, GFP-RAB7a that was expressed with a lentivirus in RAB7a KO cells localized to vesicles but also to tubular networks that partially co-localized with MitoTracker Red in living HeLa cells (Fig EV1G and Movie EV1), which rules out that fixation artifacts caused the mitochondria and ER localization of RAB7. As RAB7 appeared to completely shift to LAMP-decorated vesicles upon loss of retromer, we next tested whether endogenous RAB7 was still present on other endomembranes in those cells. Co-staining of endogenous RAB7 with TOM20 and a fluorescent ER marker in retromer-deficient cells revealed that RAB7a no longer localized to mitochondria and the ER in these cells (Figs 1C and EV1H). The loss of RAB7a from mitochondria in retromer null cells was also confirmed biochemically by pulling down mitochondria with a mito-chondrially anchored GFP tag from detergent-free lysates (Fig 1D) as well as with a commercial mitochondria immunoisolation kit (Fig EV2A). We concluded that the loss of essential retromer subunits VPS29 and VPS35 led to a striking shift in RAB7a localization from abundant endomembranes such as the ER, the TGN, and mitochondria to the late endosomal/lysosomal network.

### Loss of retromer causes hyperactivation of RAB7

Since we suspected that this shift in localization reflected changes in the activity status of the RAB7a GTPase, we next investigated this directly. We first tested how inactive, GDP-locked GFP-RAB7-T22N and constitutively active GFP-RAB7-Q67L related to lysosomes as well as mitochondria and ER membranes. Clearly, inactive GFP-Rab7 localized almost fully to ER-marker- and TOM20-positive membranes and was largely absent from lysosomes, whereas the GTP-locked mutant localized exclusively to a vesicular compartment that was positive for the lysosomal marker LAMP2 (Figs 2A and EV2B–D), indicating that activity state controls the localization of

**Figure 1. Loss of retromer leads to a pronounced shift in RAB7 distribution.**

A   Parental HeLa cells, two clonal VPS35 knockout cell lines, and one clonal VPS29 KO cell line were fixed in PFA and co-stained for endogenous RAB7a (green) and endogenous LAMP2 (red). Co-localization was analyzed across three independent experiments.

B   A clonal RAB7a knockout cell line was mixed 1:1 with parental HeLa cells and seeded onto coverslips. Following PFA fixation, the mixed cells were stained for endogenous RAB7a (green) and endogenous LAMP2 (red). Note that the RAB7a signal completely disappears in the cells not expressing RAB7a.

C   Parental HeLa cells and clonal VPS35 KO cells were co-stained for endogenous RAB7a (green) and endogenous TOM20 (red, upper panel) or for endogenous RAB7a and a mCherry-tagged ER marker (red, lower panel), and co-localization between RAB7a and the respective marker was analyzed across two independent experiments. To show that RAB7a localizes to the ER and mitochondria, endogenous TOM20 (blue) was co-stained in the lower panel.

D   Parental HeLa cells and clonal VPS35 and VPS29 KO cells were transduced with a lentivirus expressing GFP-FIS1TM (eGFP with a C-terminal mitochondrial targeting sequence and transmembrane domain of the mitochondrial protein FIS1) and disrupted through a fine needle in detergent-free sucrose buffer followed by isolation of the mitochondria from postnuclear supernatants with GFP-trap agarose beads. The amount of RAB7 precipitating with the mitochondria was quantified over four independent experiments.

Data information: All scale bars = 10 μm, all error bars = SD, and *P < 0.05 in a *t*-test of the respective condition compared to the control cells.
Source data are available online for this figure.

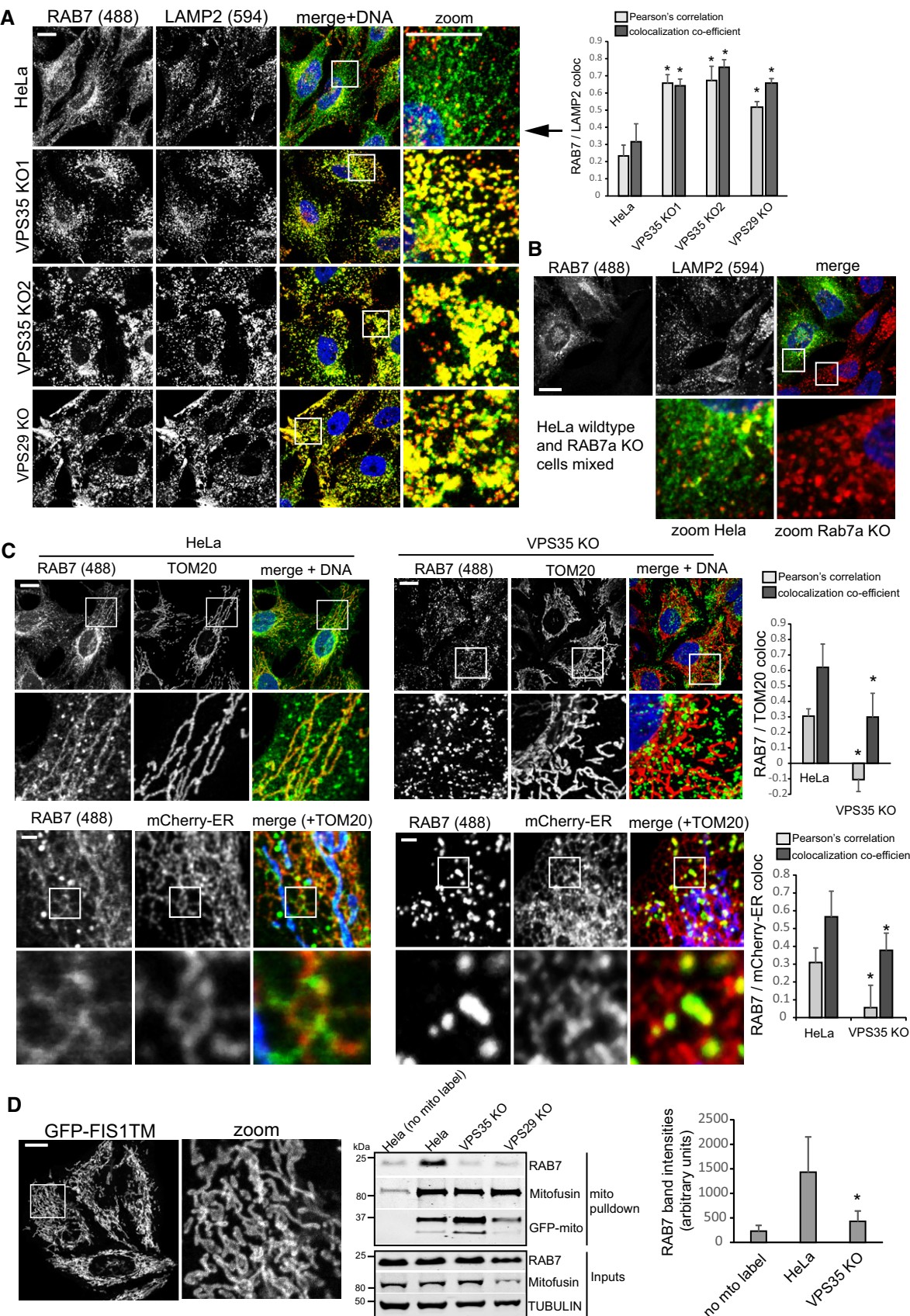

Figure 1.

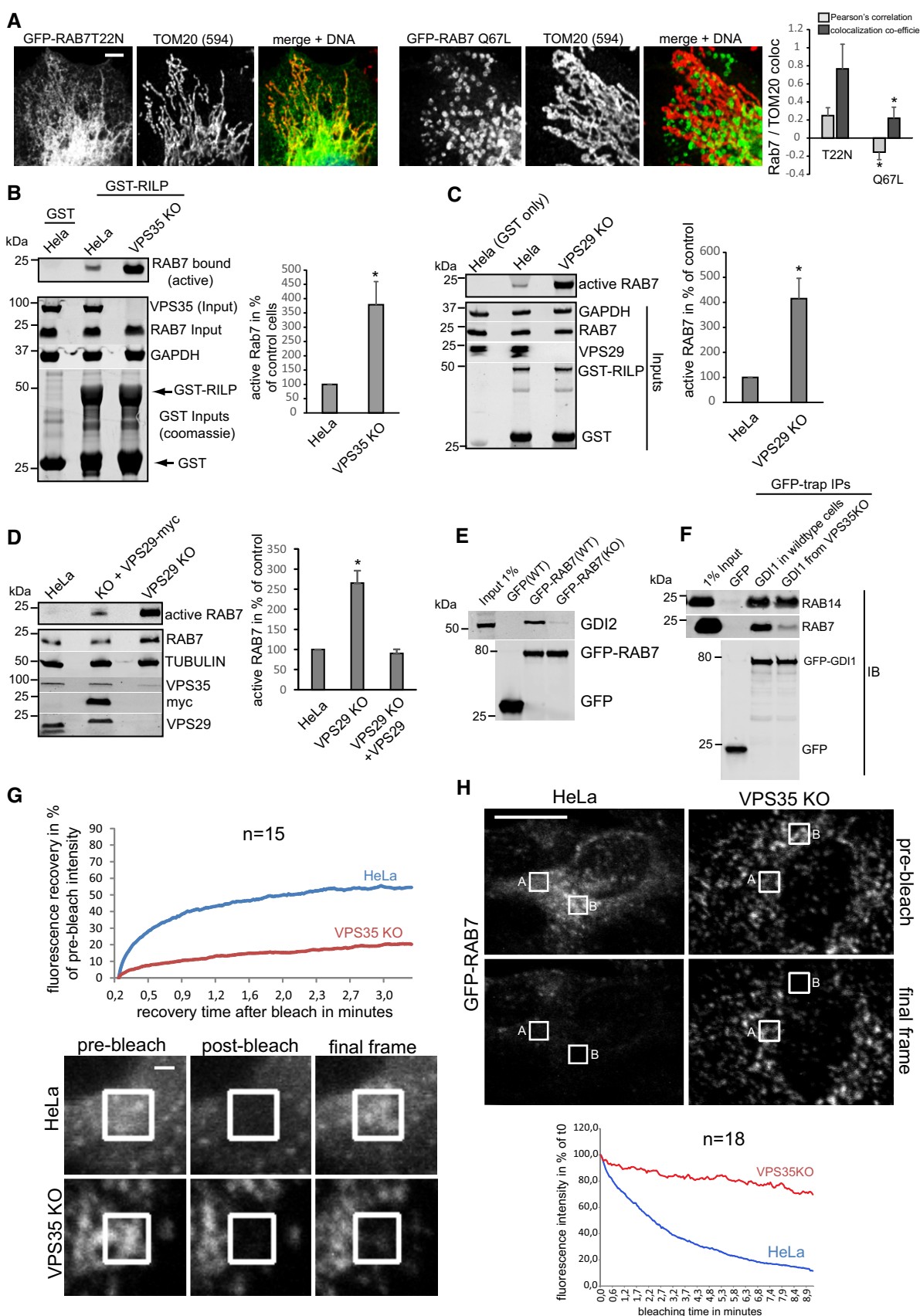

Figure 2.

◄

**Figure 2.  Retromer controls RAB7 activity levels and mobility/membrane turnover.**

A   GDP-locked (inactive) GFP-RAB7-T22N and GTP-locked (constitutively active) GFP-RAB7-Q67L were lentivirally expressed in RAB7 KO cells and co-stained with the mitochondrial marker TOM20 (red). Co-localization was analyzed over two independent experiments with 10 images each.

B   Lysates from parental HeLa cells and clonal VPS35 KO cells were probed with immobilized GST-RILP protein, and GST-RILP-bound (active) RAB7a was detected and quantified by fluorescent Western blotting across three independent experiments.

C   Lysates from parental HeLa cells and clonal VPS29 KO cells were probed with immobilized GST-RILP protein, and GST-RILP-bound (active) RAB7a was detected and quantified by fluorescent Western blotting across three independent experiments.

D   Lysates from parental HeLa cells and clonal VPS29 KO cells and VPS29 KO cells with lentivirally re-expressed VPS29-myc were probed with immobilized GST-RILP protein, and GST-RILP-bound (active) RAB7a was detected and quantified by fluorescent Western blotting across three independent experiments.

E   Lentivirally expressed GFP-RAB7 was precipitated from parental and from VPS35 KO cells, and the precipitates were analyzed for the presence of the endogenous RAB-chaperone GDI2.

F   Lentivirally expressed GFP-GDI1 was precipitated from parental and from VPS35 KO cells, and the precipitates were analyzed for the presence of endogenous RAB14 and endogenous RAB7a.

G   GFP-RAB7 was transduced into parental HeLa cells and VPS35 KO cells and analyzed for its mobility/membrane turnover using FRAP imaging in live cells. The recovery kinetics were obtained by averaging 15 FRAP recoveries acquired in two independent experiments.

H   GFP-RAB7 was transduced into parental HeLa cells and VPS35 KO cells and analyzed for its mobility/membrane turnover using FLIP imaging in live cells. The depletion kinetics (in area A, as indicated) were obtained by averaging 18 FLIP depletions acquired in two independent experiments.

Data information: All scale bars = 10 μm, all error bars = SD, and *$P < 0.05$ in a *t*-test of the respective condition compared to the control cells.
Source data are available online for this figure.

this GTPase. This was also confirmed in living cells, again ruling out fixation artifacts (Fig EV2E, and Movies EV2 and EV3). If retromer really controlled RAB7 activity state, the RAB7 mutants that are locked into their respective states should not be affected by loss of retromer. Indeed, inactive GFP-RAB7-T22N and the constitutively active GFP-RAB7-Q67L displayed no changes in their localization when expressed in parental and in VPS35-deficient cells (Fig EV3A and B). Using an established RAB7 effector pulldown assay based on the affinity of active, GTP-loaded RAB7 for its lysosomal effector RILP (Sun *et al*, 2009), we found that much more active RAB7a bound to the GST-RILP probe in lysates from VPS35-deficient cells (Fig 2B), which could be reverted by lentiviral re-expression of VPS35 (Fig EV3C). We also detected a similar increase in active RAB7 in VPS29-deficient cells (Fig 2C), which could also be rescued by lentiviral re-expression of VPS29-myc (Figs 2D and EV3D). It should be noted here that the RILP assays need to be viewed with some caution as retromer is a RAB7 effector itself (Rojas *et al*, 2008; Seaman *et al*, 2009; Harrison *et al*, 2014), so that the changes in RILP-bound RAB7 could be partially caused by less competition between effectors in the lysate. However, spiking of recombinant VPSS35, which is the direct binding partner of RAB7 (Harrison *et al*, 2014), into the lysate before RILP-bead addition had no effect on the amount of active RAB7 binding to the beads (Fig EV3E). Nevertheless, we aimed to assess the activity status by other means to confirm the hyperactivated state of RAB7 upon loss of retromer. To this end, we performed two runs of comparative quantitative proteomics with swapped SILAC labels on GFP-RAB7a expressed in wild-type HeLa cells and in VPS35 KO cells (Fig EV3F). This revealed that GFP-RAB7 bound more to its endogenous effector RILP in the VPS35 KOs, confirming our data. It also revealed a drastic (5- to 12-fold) reduction in binding to the RAB chaperones PRA-1, GDI1, and GDI2, all of which only bind to RAB GTPases in their inactive state (Rak *et al*, 2003; Sivars *et al*, 2003; Seabra & Wasmeier, 2004) (Fig EV3F). This loss of binding was confirmed by Western blotting of GFP-RAB7 IPs for endogenous GDI2, which no longer co-precipitated with GFP-RAB7 in VPS35 KO cells (Fig 2E). Similarly, GFP-trap isolation of lentivirally expressed GFP-GDI1 revealed a pronounced reduction in the binding to endogenous RAB7a in VPS35-deficient cells but had no effect on binding of endogenous RAB14 (Fig 2F), further confirming that there is much

less inactive RAB7 in those cells. Finally, we tested whether the changes in RAB7 activity levels in retromer-deficient cells result in a similar loss of RAB7 mobility and membrane turnover as has been reported for the GTP-locked RAB7-Q67L mutant (McCray *et al*, 2010). Strikingly, we found that the GFP-RAB7 signal in parental HeLa cells rapidly recovered after photobleaching in FRAP (fluorescence recovery after photobleaching) assays, whereas the GFP-RAB7 signal in VPS35 KOs barely recovered at all (Fig 2G). To exclude the possibility that the impaired FRAP recovery rates merely reflected local changes in RAB7 dynamics, we also performed FLIP (fluorescence loss in photobleaching) assays to measure GFP-RAB7 turnover within an entire cell. The FLIP assays revealed that within 9 min, nearly all cellular GFP-RAB7 signal was lost due to continuous bleaching of a small area in parental HeLa cells, whereas there was almost no detectable loss of GFP-RAB7 signal outside of the bleaching area in VPS35 KO cells (Fig 2H). Overall, our data suggest that upon loss of retromer, RAB7 activity state is strongly shifted toward the active state, which leads to accumulation of active RAB7 on late endosomes/lysosomes and to a striking loss of RAB7 mobility/membrane turnover in these cells.

**RAB7 and retromer localize to the endo-lysosomal interface**

In order to modulate RAB7 activity and prevent lysosomal accumulation, retromer should co-localize with RAB7 near lysosomal membranes. Thus, we next tested how retromer and RAB7 spatially relate to each other and relative to lysosomes in wild-type HeLa cells. To do so, we made use of the observation that methanol fixation accentuated the vesicular pool of RAB7 over the inactive population on mitochondria and the ER when imaged at low-laser settings, which made visualization of any co-localization much more accessible than in the paraformaldehyde-fixed cells (Fig EV3G). Similar to what has been reported for GFP-RAB7 and VPS26 (Rojas *et al*, 2008), our imaging revealed extensive co-localization of the retromer components SNX1 and VPS35 with endogenous RAB7 in discrete and punctate endosomal microdomains, which were often found immediately adjacent to LAMP2-decorated late endosomes/lysosomes (Fig EV4A). VPS35 in turn localizes to EEA1-decorated endosomes, albeit in a juxtaposed subdomain (Fig EV4A). Interestingly, we detected far more spatial separation

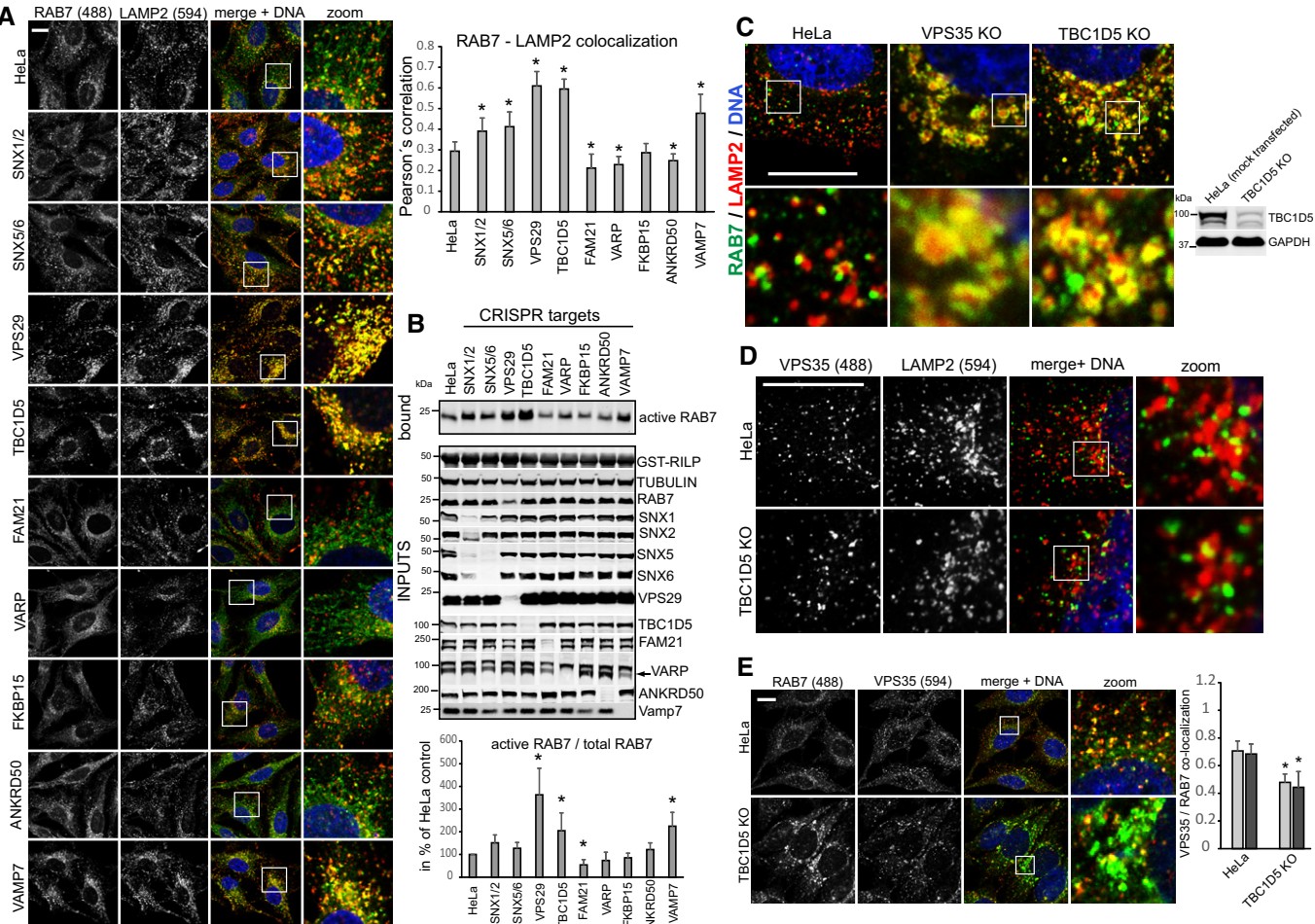

**Figure 3. A CRISPR/Cas9 screen identifies TBC1D5 as the retromer-associated component that controls RAB7 activity and localization.**

A   HeLa cells were co-transfected with CRISPR/Cas9 constructs targeting the indicated genes and a puromycin resistance marker. Five days after puromycin selection of transfected cells, all cell lines were stained for endogenous RAB7a and the lysosomal marker LAMP2, and co-localization was quantified across 12 images acquired in two independent experiments.

B   The cell lines described above for panel (A) were lysed, and RAB7a activity was assayed with the GST-RILP activity assay. Western blotting was used to confirm the efficiency of the CRISPR/CAS9 targeting approach. RILP-bound RAB7a was quantified over three independent experiments.

C   Methanol-fixed HeLa cells and VPS35 and TBC1D5 KO cells were co-stained for endogenous RAB7a (green) and endogenous LAMP2 (red).

D   Methanol-fixed parental HeLa cells and TBC1D5 KO cells were stained for endogenous VPS35 (green) and LAMP2 (red).

E   Methanol-fixed parental HeLa and TBC1D5 KO cells were co-stained for endogenous RAB7a (green) and endogenous VPS35 (red), and co-localization was quantified over two independent experiments.

Data information: All scale bars = 10 μm, all error bars = SD, and *$P < 0.05$ in a *t*-test of the respective condition compared to the control cells.
Source data are available online for this figure.

between lysobisphosphatidic acid (LBPA)-labeled late endosomes and VPS35 than seen with VPS35 and LAMP2, indicating that retromer (and thus RAB7) is more closely associated with lysosomes than with late endosomes (Fig EV4B). Triple imaging and 3D reconstruction of GFP-RAB5, LAMP2, and retromer and WASH complex components VPS35 and FAM21 revealed that retromer and thus also RAB7 were often localized directly between a RAB5-decorated endosomal domain and a LAMP2-positive late endosome/lysosome (Fig EV4C). Depletion of RAB7a and VPS35 from HeLa cells with siRNA led to a comparable loss of the retromer cargo GLUT1 from the cell surface, suggesting that RAB7 and retromer not only co-localize extensively but also form a functional unit (Fig EV4D). We

propose that the RAB7–retromer complex localizes to the interface between EEA1- and RAB5-positive sorting endosomes and late endosomes/lysosomes, which reconciles the reported endosomal localization of retromer with the role of RAB7 on lysosomes and in endosome-to-lysosome fusion.

### Retromer controls RAB7 activity through TBC1D5

Since retromer, at least to current knowledge, does not harbor intrinsic RAB-GTPase-activating (GAP) activity, we next performed a CRISPR/Cas9 screen with three mixed gRNAs against each gene to identify retromer-associated components that mediate the RAB7

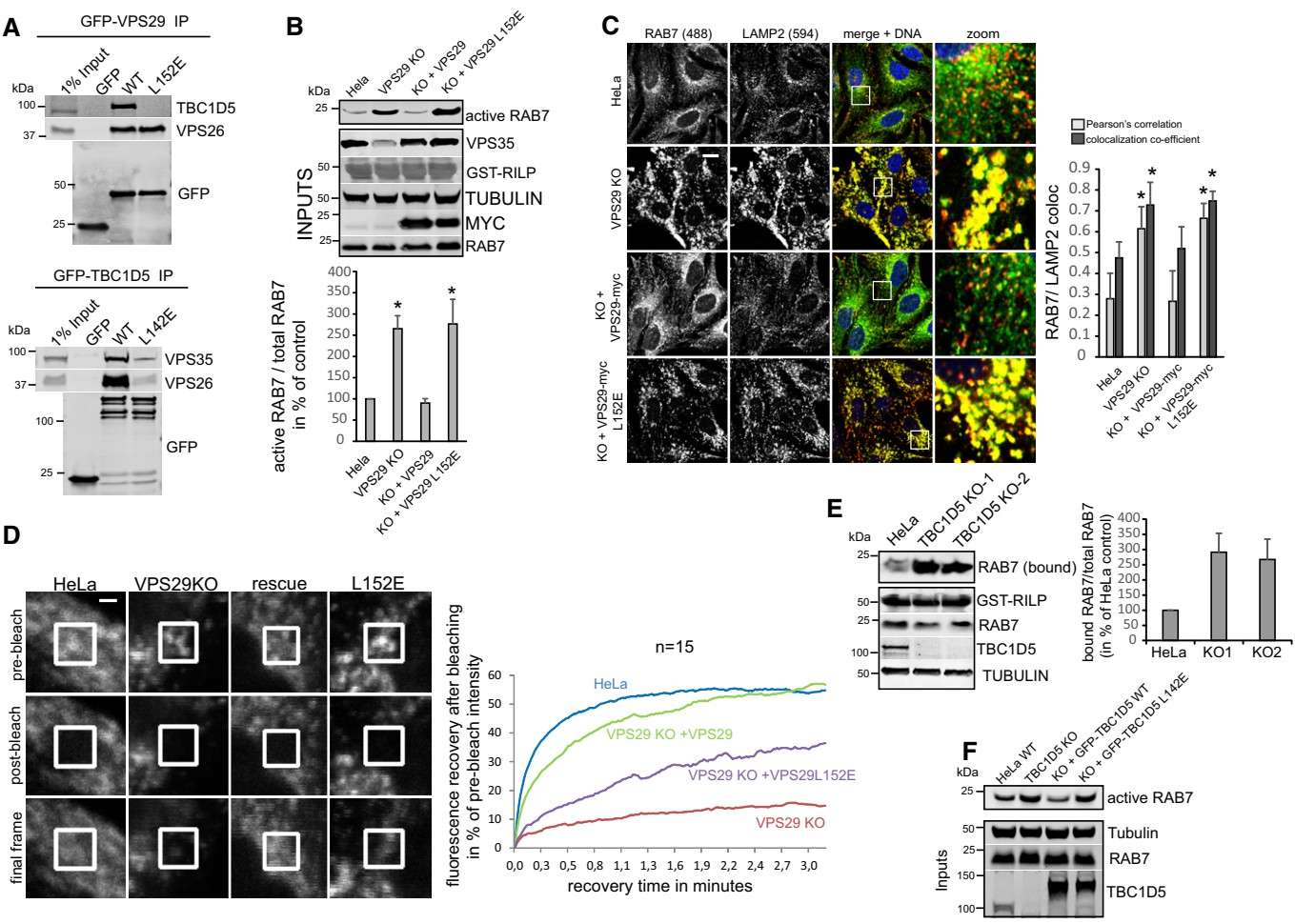

**Figure 4. TBC1D5 and retromer cooperate in the control of RAB7 activity and mobility.**

A   GFP-trap IPs of the indicated GFP-tagged VPS29 (upper panel) or TBC1D5 (lower panel) constructs confirm that the VPS29-L152E and the TBC1D5-L142E mutant lose binding to each other.

B   Parental HeLa cells, VPS29 KO cells, and VPS29 KO cells transduced with the indicated VPS29 rescue constructs were lysed, and the activity of RAB7a was analyzed with the GST-RILP assay. RAB7a activity was quantified over four independent experiments. Note that re-expression of both VPS29 variants fully restores the level of endogenous VPS35.

C   PFA-fixed parental HeLa cells, VPS29 KO cells, and VPS29 KO cells transduced with the indicated VPS29 rescue constructs were co-stained for endogenous RAB7a (green) and endogenous LAMP2 (red), and co-localization was quantified over three independent experiments.

D   Parental HeLa cells and VPS29 KO cells as well as VPS29 KO cells transduced with the indicated VPS29 rescue constructs were transduced with GFP-RAB7, and RAB7 mobility/turnover was analyzed by FRAP in living cells. The displayed recovery kinetics were obtained by averaging kinetics from fifteen FRAP recoveries per condition acquired in two independent experiments.

E   Parental HeLa cells, VPS29 KO cells, and VPS29 KO cells transduced with the indicated VPS29 rescue constructs cells were lysed, and the activity of RAB7a was analyzed with the GST-RILP assay. RAB7a activity was quantified over four independent experiments.

F   Parental HeLa cells and clonal TBC1D5 KO cells and TBC1D5 KO cells transduced with the indicated GFP-TBC1D5 rescue constructs cells were lysed, and the activity of RAB7a was analyzed with the GST-RILP assay.

Data information: All scale bars = 10 μm, all error bars = SD, and *$P < 0.05$ in a *t*-test of the respective condition compared to the control cells.
Source data are available online for this figure.

---

activity control. Our CRISPR targeting approach was highly efficient as it led to a near-complete loss of most of the targeted proteins from CRISPR-plasmid-transfected cells (Fig 3B). Confocal analysis of endogenous RAB7 and LAMP2 revealed a pronounced increase in co-localization in VPS29 CRISPR-transfected cells which were used as positive control in our screen (Fig 3A). Strikingly, knockout of TBC1D5 replicated this shift to lysosomes in full, whereas knockout of SNX1/2, SNX5/6, and the retromer-associated SNARE VAMP7

(Hesketh *et al*, 2014) partially phenocopied the increase. Loss of the retromer-associated proteins VARP (Hesketh *et al*, 2014) and FAM21 (Gomez & Billadeau, 2009) led to a decrease in lysosomal RAB7, suggesting lower RAB7 activity (Fig 3A). We could not detect significant changes with the retromer and FAM21 binding proteins ANKRD50 (Kvainickas *et al*, 2017b) and FKBP15 (Harbour *et al*, 2012). The same set of cells was also subjected to the GST-RILP-based RAB7 activity assay, which confirmed the results

from the microscopy-based screen, as loss of TBC1D5 and to a lesser extent SNX1/2 and VAMP7 led to an increase in RILP-bound, active RAB7, whereas loss of FAM21 and VARP resulted in less active RAB7 that bound to the beads (Fig 3B). Since loss of TBC1D5 appeared to fully phenocopy the loss of VPS29 or VPS35, we next investigated the role of TBC1D5 in more detail. Knockout of TBC1D5 resulted in the same loss of spatial RAB7 restriction to an endo-lyso-somal subdomain as loss of VPS35 (Fig 3C). Loss of TBC1D5 did not affect retromer localization in any obvious way (Fig 3D), but instead allowed RAB7 to leave its tight association with retromer in a spatially restricted endo-lysosomal subdomain to cover the entire lysosomal surface (Fig 3E), which also resulted in a measurable loss of co-localization between retromer and RAB7. To prove that retromer controls RAB7 activity through TBC1D5, we next generated retromer and TBC1D5 mutants that are unable to engage each other. TBC1D5 was shown to contact retromer through direct binding to the retromer subunit VPS29 with a critical involvement of the L152 residue in VPS29 (Jia *et al*, 2016). On the TBC1D5 side, residue L142 was identified as being critical for the binding to VPS29 (Jia *et al*, 2016). GFP-trap precipitations with GFP-VPS29-L152E and GFP-TBC1D5-L142E confirmed that these mutants lost binding to each other (Fig 4A), so that lentiviral rescue vectors with VPS29-myc and GFP-TBC1D5 were constructed based on these mutations. Starting with VPS29 rescues, we first tested whether the TBC1D5 binding mutant was able to rescue the increased activity of endogenous RAB7a in the GST-RILP-based effector assay. While VPS29-myc fully rescued the increased activity back to the level of control cells, VPS29-L152E did not rescue the increased activity at all (Fig 4B). Importantly, both wild-type VPS29 and the mutant VPS29 fully restored the levels of endogenous VPS35 (Fig 4B), which is degraded to a large extent upon loss of VPS29. This confirmed that the RILP effector assay really detected an increase in RAB7 activity rather than just decreased competition between the GST-RILP probe and the endogenous effector retromer. In agreement with the RILP assay, re-expression of the mutant VPS29-L152E completely failed to rescue the shift of activated RAB7 toward the lysosomal compartment, while wild-type VPS29 fully reverted it (Fig 4C). FRAP assays with GFP-RAB7 in the VPS29 KO and rescue cells revealed that the VPS29-L152E mutant only partially restored RAB7 mobility/membrane turnover, while wild-type VPS29 did so efficiently (Fig 4D). Turning toward TBC1D5, we first tested whether two clonal TBC1D5 KO cell lines also displayed increased RAB7 activity with the RILP effector assay. Both TBC1D5 KO clones displayed strongly enhanced RAB7 binding to the probe similar to the effects seen with the mixed population of TBC1D5 KOs that were used for the screen (Fig 4E). Re-expression of GFP-TBC1D5 but not of GFP-TBC1D5-L142E restored the level of active RAB7 back to control levels (Fig 4F), thereby confirming that TBC1D5 needs to engage retromer to modulate RAB7 activity. We concluded that a complex of retromer and TBC1D5 is required to turn off RAB7 in order to prevent lysosomal accumulation and immobility of hyperactivated RAB7.

## Control of RAB7 activity is not required for retromer-dependent recycling

We next tested whether TBC1D5 was needed for retromer-based recycling of integral membrane proteins. The retromer-associated SNX-BAR proteins mediate the endosome to TGN retrieval of the

hydrolase receptor cation-insensitive mannose 6P receptor (CI-MPR), while the SNX27–retromer mediates the endosome-to-plasma membrane sorting of the glucose transporter GLUT1 (Seaman, 2004; Steinberg *et al*, 2014). Loss of TBC1D5 was recently shown to impact upon retromer-dependent recycling of integrin alpha 5 and CI-MPR (Jia *et al*, 2016). To our surprise, we could not confirm an essential role of TBC1D5 in retromer-based recycling. Both CI-MPR localization at the TGN and GLUT1 localization at the plasma membrane were unperturbed in TBC1D5 knockout cells, while loss of VPS29 or the retromer-associated sorting nexins SNX5 and SNX6 resulted in pronounced sorting defects (Fig EV5A and B). In agreement with this, the TBC1D5 binding mutant VPS29 L152E was as efficient in rescuing GLUT1 lysosomal mis-sorting as the wild-type protein (Fig 5A) and also restored GLUT1 surface levels as efficiently as wild-type VPS29 (Fig 5B). To further verify that control of RAB7 activity is dispensable for retromer-mediated recycling, we next disrupted RAB7A expression with CRISPR/Cas9 and re-expressed GFP-RAB7, the GDP-locked GFP-RAB7-T22N, and the constitutively active GFP-RAB7-Q67L at endogenous expression levels (Fig EV5C) and tested their efficiency at restoring sorting of retromer cargoes. As expected, GLUT1 lost its plasma membrane localization in RAB7A-deficient cells to accumulate in lysosomes (Fig 5C). Re-expression of GFP-RAB7a but not of inactive RAB7a T22N fully reverted this phenotype. Clearly, re-expression of RAB7a Q67L was as efficient as the wild-type protein in restoring GLUT1 localization (Fig 6C). Loss of RAB7a also led to severe dispersal of CI-MPR away from the TGN or to general loss of the protein, probably due to enhanced lysosomal turnover (Fig 5D). While re-expression of GFP-RAB7-T22N failed to rescue this CI-MPR sorting defect, GFP-RAB7 and GFP-RAB7-Q67L fully restored TGN localization of CI-MPR, thereby confirming that nucleotide cycling of RAB7 is not needed for retromer-dependent sorting. It should be noted here that these experiments also confirmed the TGN localization of RAB7 as a substantial fraction of GFP-RAB7 (but not RAB7-Q67L) was often found to localize to the TGN (Fig 5D).

## Retromer and RAB7 nucleotide cycling are required for efficient mitophagy

RAB7 and the RAB7-specific mitochondrial GAP TBC1D15 have recently been linked to the control of mitophagosome formation during Parkin-mediated mitophagy (Yamano *et al*, 2014). Given that nearly all cellular RAB7 appeared to be sequestered on lysosomes upon loss of retromer, we asked whether RAB7-dependent mito-phagy was impacted by this. We first tested whether general auto-phagy under starvation conditions was affected by loss of retromer. In agreement with published work (Orsi *et al*, 2012; Zavodszky *et al*, 2014), we detected only minor defects in autophagic flux in nutrient-starved retromer knockout cells, arguing against a major role for retromer in general autophagy (Fig EV5D). To specifically trigger mitophagy, we lentivirally expressed mCherry-tagged Parkin, an ubiquitin ligase that is recruited to damaged mitochondria (Narendra *et al*, 2008; Matsuda *et al*, 2010), and induced mitochon-drial depolarization with the proton uncoupler CCCP. While Parkin expression and mitochondrial morphology were comparable across parental HeLa cells and VPS29, VPS35, and TBC1D5 KO cells, CCCP-induced clearance of mitochondrial mass (as evidenced by residual TOM20) was severely perturbed in all three mutants

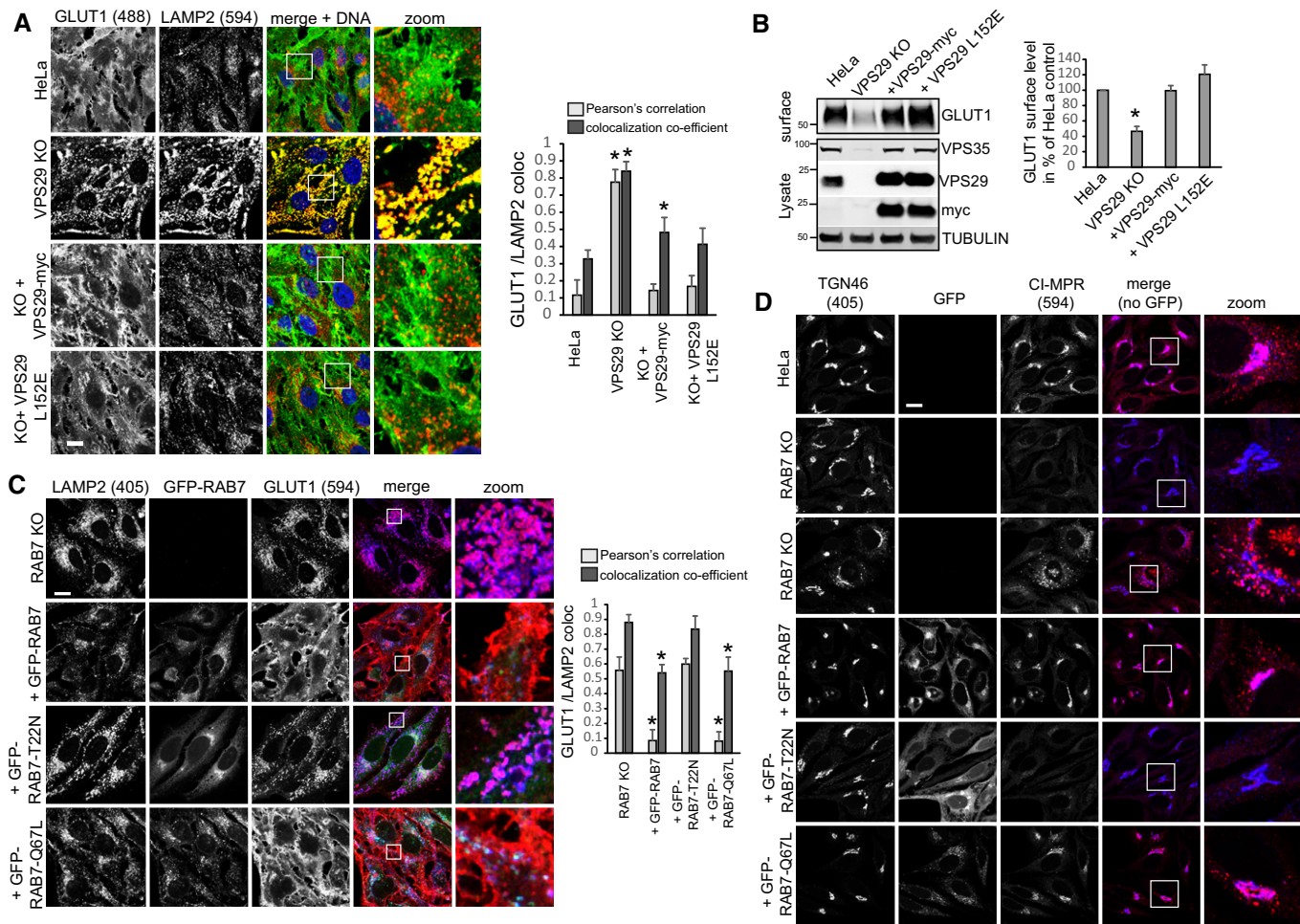

**Figure 5. Control of RAB7 activity is not required for retromer-based sorting of integral membrane proteins.**

All images show formaldehyde-fixed cells.

A   Parental HeLa cells, VPS29 KO cells, and VPS29 KO cells transduced with the indicated VPS29 rescue constructs cells were co-stained for endogenous GLUT1 (green) and endogenous LAMP2 (red), and co-localization was quantified over three independent experiments.

B   Parental HeLa cells, VPS29 KO cells, and VPS29 KO cells transduced with the indicated VPS29 rescue constructs were surface-biotinylated, followed by streptavidin isolation and Western blot-based quantification of biotinylated surface proteins. Surface GLUT1 was quantified over four independent experiments.

C   RAB7a knockout cells and RAB7 KO cells transduced with the indicated GFP-RAB7 rescue constructs cells were co-stained for endogenous GLUT1 (red) and endogenous LAMP2 (blue), and co-localization was quantified over two independent experiments.

D   Parental HeLa cells, RAB7a knockout cells, and RAB7 KO cells transduced with the indicated GFP-RAB7 rescue constructs were co-stained for endogenous CI-MPR (red) and endogenous TGN46 (blue).

Data information: All scale bars = 10 μm, all error bars = SD, and *$P < 0.05$ in a *t*-test of the respective condition compared to the control cells.
Source data are available online for this figure.

(Fig 6A and B). The clearance of mitochondria in VPS29 KO cells could be fully rescued by re-expression of VPS29 but not by VPS29-L152E, indicating that the control of RAB7 activity is needed for efficient mitophagy (Fig 6A and B). Similarly, mutant GFP-TBC1D5-L142E also failed to rescue TOM20 clearance, while wild-type GFP-TBC1D5 rescued efficiently (Fig EV6A). To test whether a similar phenotype could also be observed under more physiological conditions, we employed SHSY-5Y cells which express Parkin endogenously (Bouman *et al*, 2011). To visualize the much more subtle mitophagy without overexpressed Parkin, we transduced these cells with a sensitive mitophagy sensor based on a tandem mCherry-GFP-FIS1TM construct that is anchored in the

mitochondrial membrane and undergoes a red shift due to lysosomal quenching of the acid-sensitive GFP moiety once a mitochondrion is delivered to a lysosome through mitophagy (Allen *et al*, 2013). Preliminary tests with this construct in HeLa cells stably expressing HA-Parkin confirmed that it is indeed an excellent sensor for mitophagy, as we detected almost no red-shifting in control cells, whereas cells treated with CCCP displayed pronounced red-shifting after 6 h and complete shift to red-only label after 16 h of CCCP treatment (Fig EV6B). The SHSY-5Y cells expressing this sensor were transfected with control and with VPS35-specific siRNA and treated with the iron chelator deferiprone for 24 h, which was shown to be an inducer of mitophagy (Allen *et al*, 2013). A

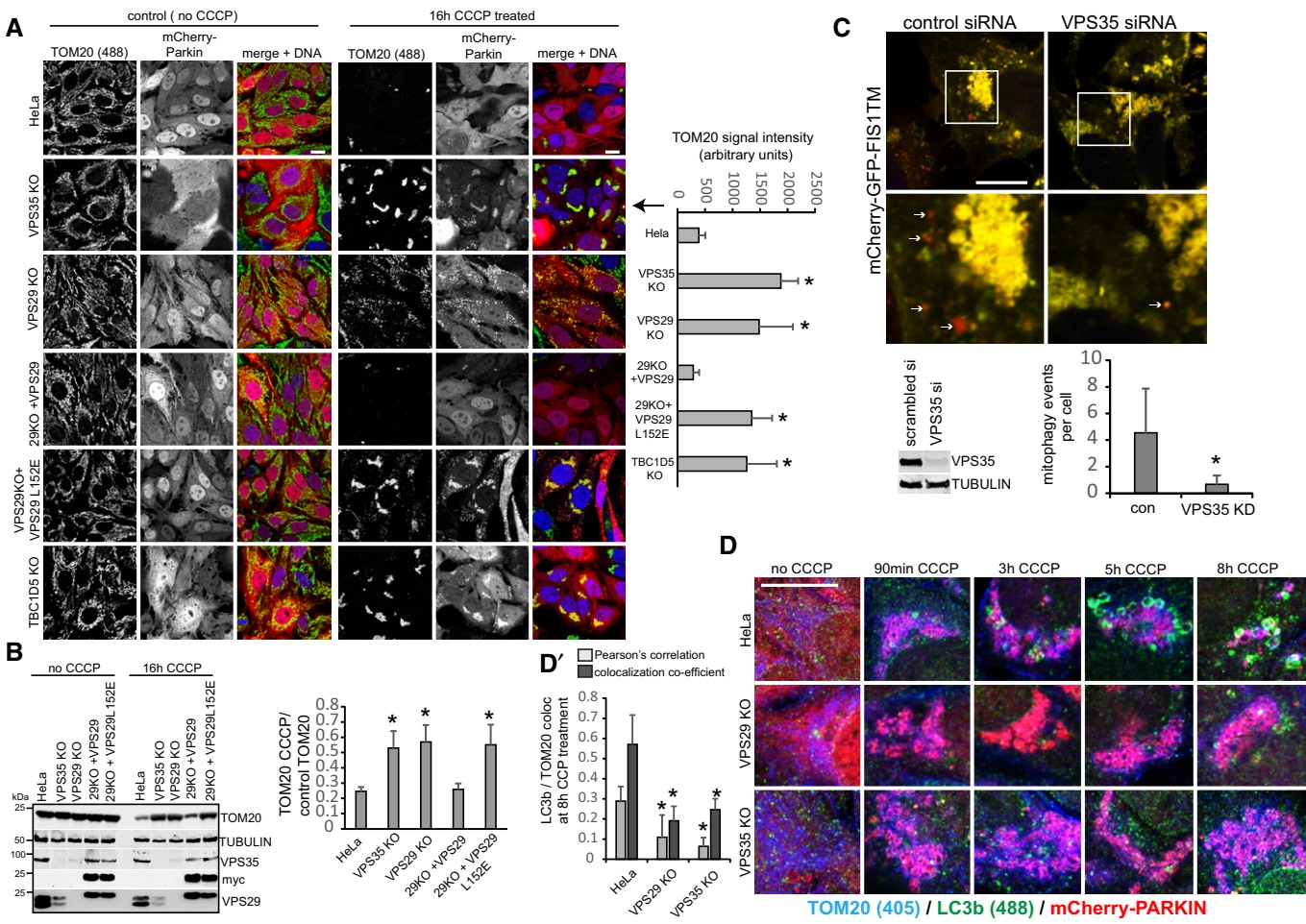

**Figure 6.   Mitophagy is defective upon loss of retromer.**

A   mCherry-Parkin-transduced parental HeLa cells, VPS35 KO cells, VPS29 KO cells, and VPS29 KO cells transduced with the indicated VPS29 rescue constructs were treated with CCCP over 16 h, followed by PFA fixation and staining of endogenous TOM20 (green). Residual TOM20 after CCCP treatment was quantified over two independent experiments.

B   The cells described for panel (A) were lysed after 16-h CCCP treatment, and residual TOM20 was detected by Western blotting (left) and quantified over three independent experiments (right).

C   SHSY-5Y cells that had been transduced with a construct expressing tandem mCherry-eGFP-FIS1TM (mitochondria-anchored mCherry-GFP tandem) were transfected with control and VPS35-specific siRNA and treated with the iron chelator deferiprone for 24 h. The cells were then assessed for red-shifted dots representing mitophagy events by confocal microscopy. Efficiency of the siRNA treatment was confirmed by Western blotting against endogenous VPS35. Mitophagy events were counted in 20 images per condition, acquired in two independent experiments.

D   Parental HeLa cells and VPS29 and VPS35 KO cells were transduced with mCherry–Parkin, and mitophagy was induced for the indicated time points with the proton uncoupler CCCP. The cells were then fixed in methanol and co-stained for endogenous LC3b (green) and TOM20 (blue). (D′) Co-localization between LC3b and TOM20 was analyzed over 12 images per condition acquired in two independent experiments.

Data information: All scale bars = 10 μm, all error bars = SD, and *$P < 0.05$ in a *t*-test of the respective condition compared to the control cells.
Source data are available online for this figure.

microscopic analysis of the number of red-shifted punctate mitophagy events revealed a significant loss of lysosomal delivery of damaged mitochondria in the VPS35-depleted cells, thereby confirming our data from HeLa cells (Fig 6C).

### Loss of retromer causes mitophagosome formation and ATG9a trafficking defects

We next asked whether the observed defect in mitochondrial clearance was due to perturbed mitophagosome formation or due to perturbed mitophagosome maturation and/or degradation. In a

time-course experiment from 90 min to 8 h after mCherry-Parkin and CCCP initiated mitophagy, endogenous LC3b was used as a marker of mitophagosome formation. While we observed an increasing number of LC3b-encapsulated mitochondria from 3 to 8 h in parental HeLa cells, we failed to detect similar mitophagosome formation in retromer-deficient cells (Fig 6D). This was also evident in a quantitative co-localization analysis of TOM20 and LC3b 8 h after CCCP induction, suggesting that mitophagosome formation was perturbed (Fig 6D). This was confirmed with lentivirally expressed GFP-LC3b, which appeared to accelerate mitophagosome formation in the wild-type cells compared to the time course

with endogenous LC3b but also indicated severely perturbed mitophagosome formation in retromer-deficient cells (Fig 7A). While we observed nearly complete packaging of the mitochondria into LC3-positive structures in the parental HeLa cells after 4 h of CCCP treatment, GFP-LC3 was still largely absent from the damaged mitochondrial mass in the VPS35 KO cells (Fig 7A). In agreement with perturbed mitophagosome formation, we also failed to observe clustering of the endogenous ATG16L1 protein, a component of the ATG8 homologue lipidation machinery (Lamb *et al*, 2013), near the damaged mitochondrial mass, whereas such clustering was readily detected in the wild-type cells starting 3 h after CCCP addition (Figs 7B and EV7A). An analysis of ULK1 recruitment, which is involved in the initiation of mitophagy (Lamb *et al*, 2013), revealed no obvious differences in ULK translocation to the damaged mitochondria in retromer-deficient cells (Fig EV7B), suggesting that the mitophagosome formation defects were caused further downstream. Since both retromer and TBC1D5 have been linked to the trafficking of the autophagic transmembrane protein ATG9a (Popovic & Dikic, 2014; Zavodszky *et al*, 2014), we also performed a time-course analysis of ATG9a localization relative to the damaged mitochondria. The monoclonal antibody used for this analysis was found to be specific as all vesicular and perinuclear signal disappeared after CRISPR/Cas9-mediated disruption of the ATG9a gene, with only minor nuclear background staining remaining (Fig 7C). Without CCCP treatment, ATG9a was found in vesicular structures and in a perinuclear area that resembled the TGN (Fig 7D). Starting at 3 h after mitophagy induction, we observed a translocation of ATG9a vesicles to the Parkin-decorated damaged mitochondria in wild-type cells which persisted over the next 5 h (Fig 7D). In contrast, ATG9a appeared to stay in a tightly defined perinuclear cluster and did not relocate to the damaged mitochondria in retromer-deficient cells (Fig 7D).

### Control of RAB7 activity is needed for ATG9a translocation and mitophagosome formation

We next tested whether the observed defects in mitophagosome formation and ATG9a translocation were caused by the changes in RAB7 activity and localization. Since RAB7 has been reported to co-localize with ATG9a (Young *et al*, 2006; Orsi *et al*, 2012) and also to play a role in autophagosome formation around damaged mitochondria (Yamano *et al*, 2014), we speculated that the lysosomal accumulation of activated RAB7 in the retromer-deficient cells prevented RAB7 from being available to drive mitophagy in the absence of retromer. Indeed, when we analyzed the localization of endogenous RAB7 in cells currently engaged in digesting damaged mitochondria, we observed clear changes in RAB7 localization. Parental HeLa cells as well as the VPS29 rescue cells displayed a fine pattern of RAB7-positive structures covering and around the mCherry-Parkin- and TOM20-labeled damaged mitochondrial clusters (Fig 8A). In contrast, we failed to detect endogenous RAB7 on the damaged mitochondria in the VPS29 and VPS35 KO cells as well as in the VPS29-L152E rescued cells. In order to prove that the mitophagy defect was due to a loss of RAB7 activity control, we expressed mCherry-Parkin in the RAB7 KO and rescue cells described above (Fig EV5C). In contrast to wild-type GFP-RAB7, the constitutively active GFP-RAB7-Q67L failed to localize to the damaged mitochondrial mass 4 h after CCCP addition (Fig 8B). Knockout of RAB7a led to a pronounced block in mitochondrial clearance, whereas parental HeLa cells had

removed all TOM20-labeled mitochondria within 16 h after CCCP addition (Fig 8C). While GFP-RAB7 fully restored the efficiency of mitophagy, the constitutively active GFP-RAB7-Q67L failed to do so, thereby proving that control of RAB7 activity/nucleotide cycling is needed to efficiently clear damaged mitochondria during Parkin-mediated mitophagy.

Finally, we aimed to test whether the hyperactivated RAB7 would also cause similar ATG9a translocation defects as we had observed in the retromer-deficient cells. To do so, we used clonal RAB7a knockout cells that had been infected with a lentivirus expressing untagged RAB7-Q67L. These and VPS35 KO cells were transduced with mCherry-Parkin, treated with CCCP for 4 h, and stained for endogenous ATG9a and the TGN marker TGN46. Whereas ATG9a was detected dispersed throughout the cytoplasm and clustered around Parkin-labeled mitochondria in parental HeLa cells, ATG9a was detected almost exclusively at the TGN46-labeled TGN in VPS35 KOs and in RAB7-Q67L-expressing cells (Fig 9A). In the same set of cells that were not treated with CCCP, we failed to detect obvious changes in ATG9a localization, which localized mainly to the TGN and to some cytoplasmic vesicles in all cell lines (Fig EV7C). We next tested whether there were changes in RAB7 and ATG9a co-localization during mitophagy. Indeed, we detected a pronounced loss of co-localization between GFP-RAB7 and endogenous ATG9a in the VPS35 KO cells both under CCCP-treated and in untreated condition (Figs 9B and EV7D). This was particularly obvious at the TGN, where GFP-RAB7 and ATG9a no longer co-localized due to lysosomal sequestration of GFP-RAB7 upon loss of retromer (Fig EV7D). Finally, we analyzed whether constitutively active RAB7-Q67L also caused mitophagosome formation defects similar to the defects caused by loss of retromer. Indeed, we observed much less endogenous LC3b around damaged mitochondria at five and 8 h after CCCP addition in the cells expressing RAB7-Q67L (Fig 9C), suggesting that the autophagosome formation defect in retromer-deficient cells is indeed caused by the hyperactivated RAB7 that is accumulated on the lysosomal compartment.

## Discussion

Here, we describe an unexpected new function of the retromer complex in the control of cellular RAB7 activity, localization, and mobility. Our data establish that a significant fraction of inactive RAB7 associates with endomembranes such as mitochondria, the ER, and also Golgi membranes, whereas active RAB7 localizes to endo-lysosomes. This localization and the balance between inactive and activated RAB7 appear to be maintained by the RAB7 effector retromer, which associates with the GTPase-activating protein TBC1D5 to prevent lysosomal accumulation of hyperactivated and thus immobilized RAB7. The localization of endogenous RAB7 to mitochondria and the ER was an unexpected observation, which was made possible by the highly sensitive and specific rabbit monoclonal antibody against RAB7a recently developed by Abcam. Previously, we had been unable to stain endogenous RAB7 because of the insufficient strength of the respective antibodies. We believe that this is the primary reason that this distribution of RAB7 was not reported previously in spite of a very substantial body of literature on this small GTPase.

While the biological reasons for RAB7 on mitochondria and other abundant endomembranes remain to be explored further, our data

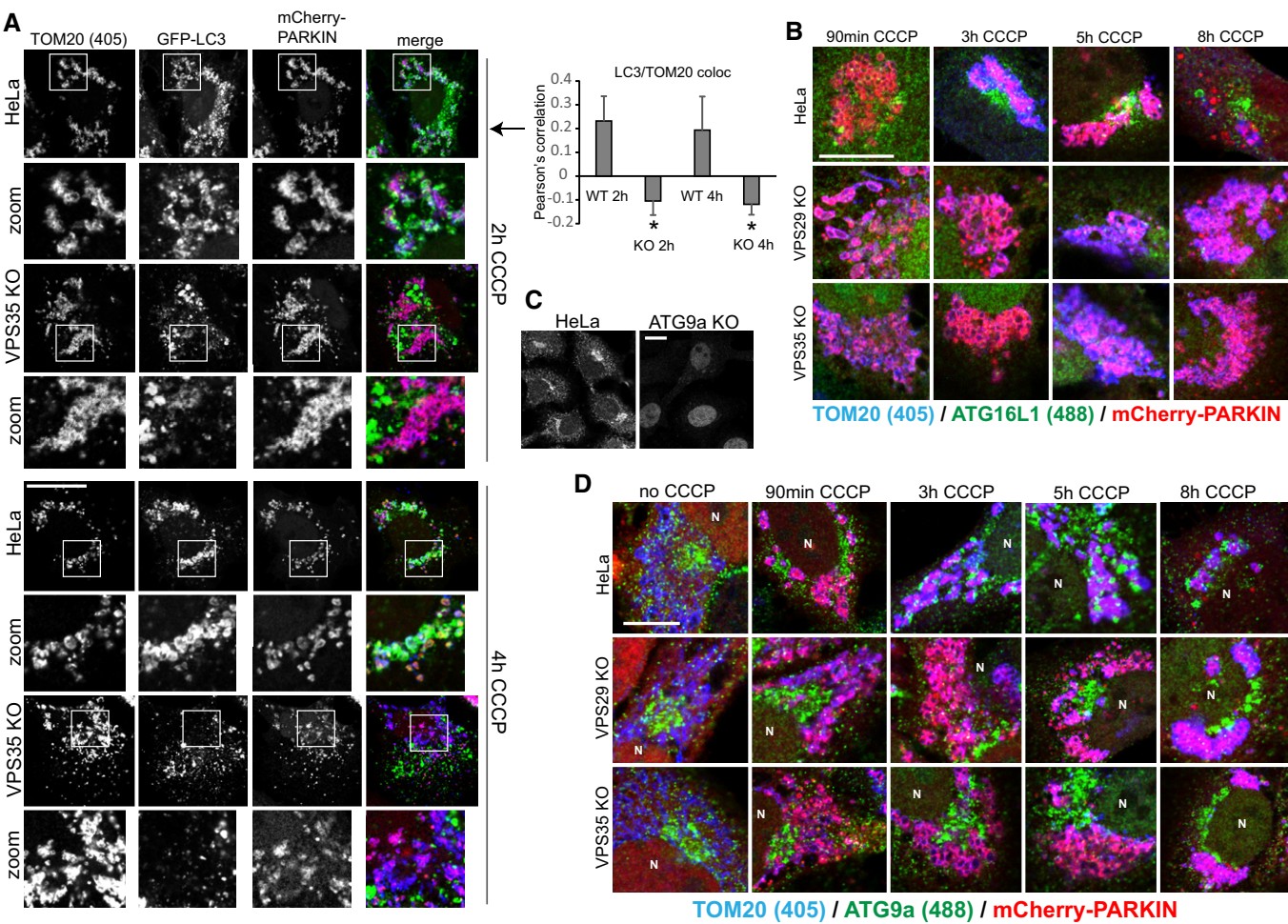

**Figure 7. Autophagosome formation and ATG9a trafficking are impaired upon loss of retromer.**

A  Parental HeLa cells and VPS35 KO cells were transduced with mCherry-Parkin and GFP-LC3b and incubated with CCCP for 2 and 4 h. Co-localization of GFP-LC3b and endogenous TOM20 (blue) was quantified over two independent experiments.

B  Parental HeLa cells and VPS29 and VPS35 KO cells transduced with mCherry-Parkin were treated with CCCP for the indicated time points, followed by staining of endogenous ATG16L1 (green) and endogenous TOM20 (blue).

C  HeLa cells were transfected with humanized Cas9 and a mix of three distinct gRNAs targeting the ATG9a gene. Note that the vesicular ATG9a antibody signal (green) completely disappears in the CRISPR-treated cells, indicating a high degree of specificity.

D  Parental HeLa cells and VPS29 and VPS35 KO cells transduced with mCherry-Parkin were treated with CCCP for the indicated time points, followed by staining of endogenous ATG9a (green) and endogenous TOM20 (blue).

Data information: All scale bars = 10 μm, all error bars = SD, and *P < 0.05 in a *t*-test of the respective condition compared to the control cells.

on retromer and its regulation of RAB7 are more straightforward. Our co-localization analysis indicated that a lot of the vesicular RAB7 appears to be localized to sharply defined, retromer-positive domains at the interface between EEA1-positive sorting endosomes and LAMP1/2-positive late endosomes/lysosomes. This localization is probably owed to the dual role of RAB7 in the recycling of trans-membrane proteins such as the CI-MPR and GLUT1 through retromer and its reported role in endo-lysosomal fusion through effectors such as the HOPS complex (Balderhaar & Ungermann, 2013). At the interface between mature endosomes and lysosomes, RAB7 could co-ordinate the recycling of membrane proteins away from the lysosomal pathway with the endo-lysosomal fusion after this recycling has taken place. Our data indicate that retromer is not only a RAB7 effector but also serves as a master regulator of RAB7

activity, localization, and mobility. Without retromer and the retromer-bound TBC1D5, RAB7 loses its sharply defined localization between endosomes and lysosomes and accumulates all over the lysosomal compartment in its active form. Constitutively active RAB7-Q67L has been shown to have dramatically reduced turnover from lysosomal membranes in FRAP assays (McCray *et al*, 2010). Our own FRAP and FLIP data suggest that RAB7 loses its membrane cycling and thus its mobility to a similarly dramatic extent, which suggests that endogenous RAB7 is essentially GTP-locked in the absence of the retromer–TBC1D5 complex. The immobilization on lysosomes caused by the loss of nucleotide cycling then results in defects in non-lysosomal functions of RAB7 due to a depletion of available pools of cycling/mobile RAB7. The extent of this pheno-type probably means that VPS29-bound TBC1D5 is the predominant

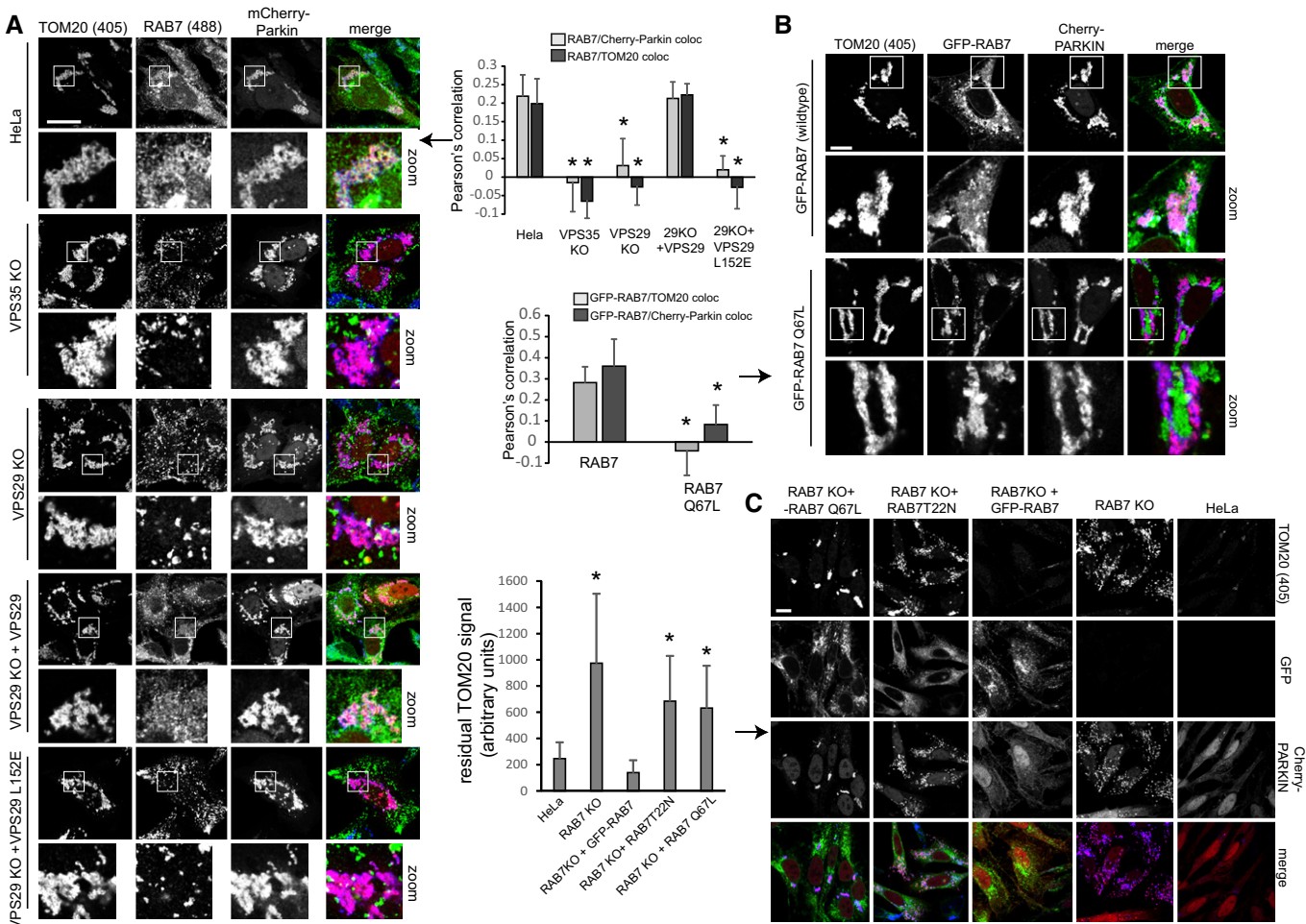

**Figure 8.  Control of RAB7 activity is required for efficient removal of mitochondria through mitophagy.**

All image panels show PFA-fixed HeLa cells.

A   mCherry-Parkin-transduced parental HeLa cells, VPS35 KO cells, VPS29 KO cells, and VPS29 KO cells transduced with the indicated VPS29 rescue constructs cells were incubated with CCCP over 4 h, followed by staining of endogenous RAB7a (green) and TOM20 (blue). Co-localization between RAB7 and TOM20 as well as RAB7 and mCherry-Parkin was quantified over two independent experiments.

B   RAB7a KO cells transduced with mCherry-Parkin and GFP-RAB7 or GFP-RAB7-Q67L were treated with CCCP for 4 h, followed by co-staining with endogenous TOM20 (blue). Co-localization between the GFP-RAB7 variants and TOM20 was quantified over two independent experiments.

C   Parental HeLa cells, RAB7 KO cells, and RAB7 KO cells transduced with the indicated GFP-RAB7 rescue constructs were treated with CCCP for 16 h, followed by staining of endogenous TOM20 (blue). The residual TOM20 signal after CCCP treatment was quantified over two independent experiments.

Data information: All scale bars = 10 μm, all error bars = SD, and *$P < 0.05$ in a *t*-test of the respective condition compared to the control cells.

mechanism to deactivate RAB7 to remove it from endo-lysosomes after cargo recycling and/or fusion of the compartment is completed.

Our data clearly indicate that the control of RAB7 activity should be considered as a novel function of the retromer complex, as it appears to be functionally separated from its well-characterized role in the recycling of transmembrane proteins. Loss of TBC1D5 alone fully reproduced the RAB7 overactivity phenotype, but left retromer localization and sorting of the retromer cargoes GLUT1 and CI-MPR completely intact. Supporting this, the TBC1D5-binding-deficient VPS29-L152E was unable to restore normal RAB7 activity and localization, but restored GLUT1 surface delivery like wild-type VPS29. This was further confirmed by our RAB7 knockout and rescue experiments, which established that GTP-locked RAB7 is fully

competent in restoring retromer-mediated recycling. We have just reported that the retrograde sorting of CI-MPR does not depend on the retromer trimer; instead, we found even slightly faster delivery of CI-MPR to the TGN in VPS35-deficient cells (Kvainickas *et al*, 2017a,b). It remains to be explored further whether the control of RAB7 activity is a modulating factor for the SNX-BAR-dependent transport of CI-MPR that we observed.

Unlike retromer-based recycling, our study establishes that the removal of damaged mitochondria through Parkin-mediated mito-phagy requires available pools of inactive RAB7 that are maintained by retromer and TBC1D5. According to our and previously published data, we propose that RAB7 likely has several distinct functions during mitophagy. Our data on mitochondrial clearance in

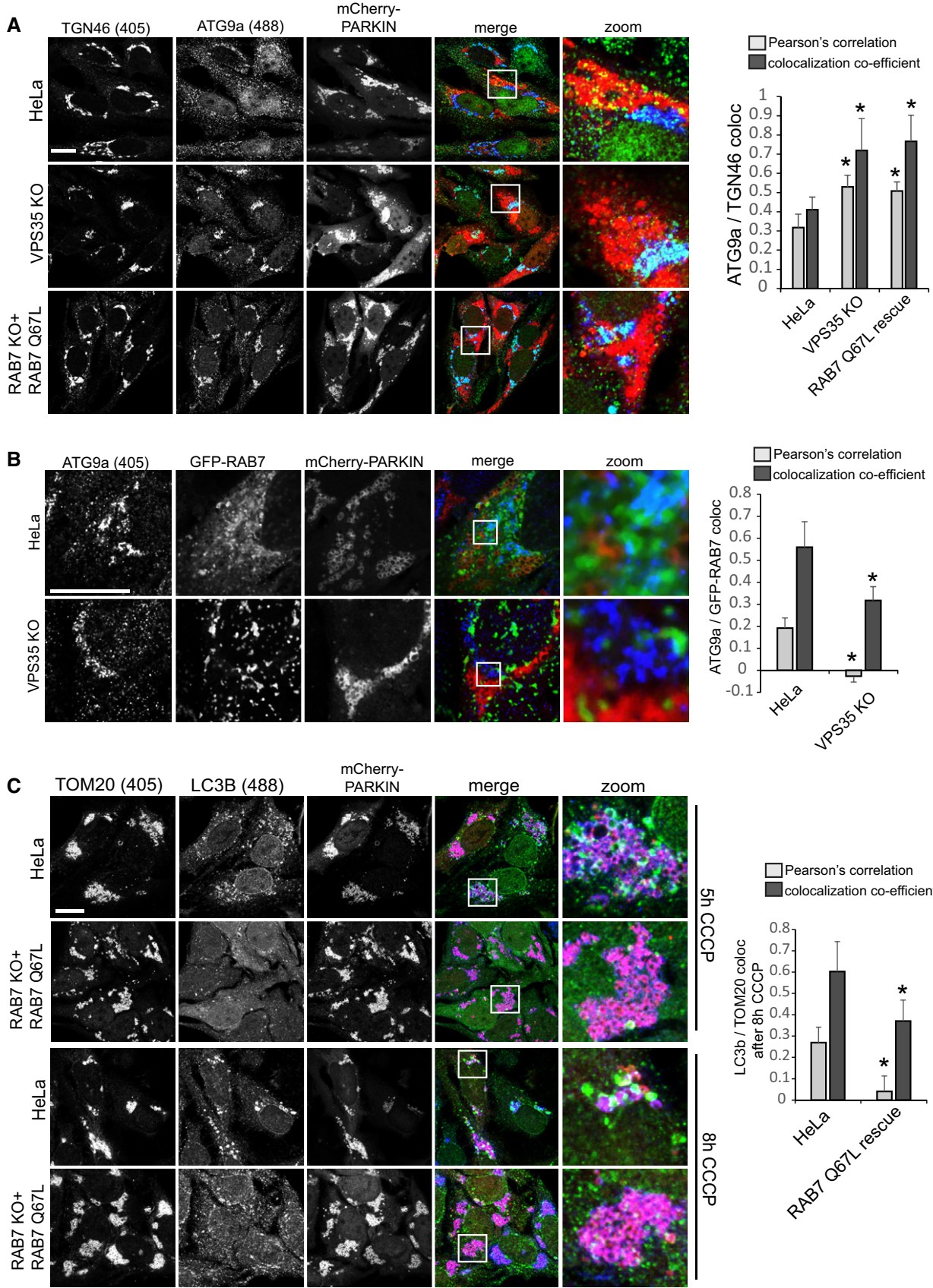

**Figure 9.**

◄

**Figure 9.  Hyperactive RAB7 results in ATG9a trafficking and autophagosome formation defects.**

A   mCherry-Parkin-expressing parental HeLa cells, VPS35 KO cells, and RAB7 KO cells transduced with untagged RAB7-Q67L were treated with CCCP for 5 h, fixed in PFA, and co-stained for endogenous ATG9a (green) and the TGN marker TGN46 (blue). Co-localization between ATG9a and TGN46 was analyzed across 14 images acquired in two independent experiments.

B   Parental HeLa cells and VPS35 KO cells were transduced with GFP-RAB7 and mCherry-Parkin and treated with CCCP for 5 h. The cells were then fixed in PFA and stained for endogenous ATG9a (blue). Co-localization between GFP-RAB7 and ATG9a was quantified across 14 images acquired in two independent experiments.

C   mCherry-Parkin-expressing parental HeLa cells and RAB7 KO cells transduced with untagged RAB7-Q67L were treated with CCCP for 5 h (upper panel) and 8 h (lower panel), fixed in methanol, and stained for endogenous TOM20 (blue) and endogenous LC3b (green). Co-localization between LC3b and TOM20 was quantified across 14 images acquired in two independent experiments.

Data information: All scale bars = 10 μm, all error bars = SD, and *$P < 0.05$ in a *t*-test of the respective condition compared to the control cells.

a RAB7a knockout background suggest that RAB7a may be involved in mitochondrial clustering upon CCCP-induced damage (Fig 8C), which appeared defective in the knockout but was restored with the RAB7-Q67L mutant. Further, the autophagic transmembrane protein ATG9a has been observed to localize to RAB7-, CI-MPR-, and retromer-decorated endosomal entities (Orsi *et al*, 2012). We observed that the co-localization between RAB7 and ATG9a was diminished upon loss of retromer as nearly all RAB7 was sequestered to a lysosomal compartment. In the absence of retromer as well as in the presence of hyperactive RAB7-Q67L, ATG9a appeared unable to leave the TGN during mitophagy, which could be explained in two distinct ways: RAB7 could directly mediate transport from the TGN to peripheral endosomes, so that its loss from ATG9a containing TGN structures due to lysosomal accumulation of RAB7 would lead to a disruption of this transport. Alternatively, RAB7 could be involved in the regulated transport of ATG9a from endosomes to the TGN. Upon loss of retromer, RAB7 can no longer regulate this transport as it is sequestered on lysosomes, so that ATG9a could be shuttled back to the TGN in an unregulated manner that does not allow for the required residence time around damaged mitochondria to promote mitophagosome formation. Tooze and colleagues did not detect any role of retromer in ATG9a trafficking in experiments that were based upon the suppression of VPS26a by siRNA (Orsi *et al*, 2012). We have recently shown for retromer-dependent GLUT1 recycling that both VPS26a and the paralogue VPS26b need to be suppressed to achieve a loss of retromer function (Gallon *et al*, 2014), which likely explains the different observations. Rubinzstein and colleagues have reported an accumulation of ATG9a at the TGN during autophagy upon suppression of endogenous VPS35 followed by re-expression of the Parkinson-related mutant VPS35-D620N (Zavodszky *et al*, 2014). We assume that re-expression of the mutant in a knockdown background has caused the full loss of retromer function that we obtained with the CRISPR/Cas9-mediated deletions so that a comparable ATG9a phenotype became apparent. Our data add the mechanistic insight into this phenotype, as ATG9a is likely not a direct retromer cargo but instead relies on RAB7-dependent transport that becomes deregulated upon loss of retromer due to RAB7 not being available for this pathway. We propose that the perturbed ATG9a trafficking could at least partially explain the pronounced autophagosome formation defects that we observed upon loss of retromer. Beyond ATG9a trafficking, RAB7 appears to also have a more direct role in mitophagosome formation. We observed that RAB7 finely covers the damaged mitochondrial structures, and others have reported that loss of the mitochondrially anchored RAB7 GAP TBC1D15 results in aberrant mitophagosome formation that is caused by deregulated RAB7 (Yamano *et al*, 2014). Our localization data indicate that RAB7 is no longer present on and around the damaged mitochondria in retromer-deficient cells, so that this role of RAB7 in mitophagosome formation is likely also perturbed upon loss of retromer. In this context, we would like to point out that TBC1D5 and the mitochondria-anchored RAB7-specific GAP TBC1D15 (Yamano *et al*, 2014) likely act in separate pathways. TBC1D5 is only active when it is bound to retromer on endo-lysosomes (Jia *et al*, 2016), a mechanism which is needed to release RAB7 from this compartment so that it is available in its inactive form to become locally activated elsewhere. TBC1D15 is strictly localized to the outer mitochondrial membrane through complex formation with the mitochondrial membrane protein FIS1 (Onoue *et al*, 2013), where it controls RAB7 activity specifically during mitophagy. Overall, we propose that retromer and TBC1D5 maintain pools of inactive and mobile RAB7 which are needed for the non-lysosomal functions of this small GTPase such as mitophagy. It remains to be investigated whether other functions of RAB7, such as endosome maturation, are also perturbed upon loss of retromer or TBC1D5.

Our findings may also be relevant from a medical perspective as mutations in the retromer complex cause hereditary Parkinsonism, a disease that has been linked to defective mitophagy (Gan-Or *et al*, 2015). In light of this, it will be of great importance to investigate whether these mutations impact upon control of RAB7 activity and thus influence the efficiency of mitochondrial clearance. LRRK1 was shown to influence the activity of RAB7 through a VAMP7- and TBC1D2-dependent mechanism (Toyofuku *et al*, 2015). Similarly, LRRK2, which is the most frequent genetic cause of Parkinson's disease, also negatively regulates RAB7 activity, and the mutations in LRRK2 that cause Parkinson's disease deactivate RAB7 excessively (Gomez-Suaga *et al*, 2014). Our CRISPR screen confirmed a role of VAMP7 in the control of RAB7 activity, which likely constitutes a parallel pathway to retromer–TBC1D5. Since retromer and LRRK2 have been shown to interact genetically, it is conceivable that the control of RAB7 activity is the common molecular target that these genetic interactions are based on. Overall, our study identified a new and unanticipated role of the retromer complex in the control of RAB7 activity and mitophagy that will certainly be the basis for further productive research.

## Materials and Methods

### Antibodies

Antibodies used in this study were as follows rabbit LC3b (3868), rabbit ATG16L1 (8089), and rabbit EEA1 (3288) from Cell Signaling Technologies; rabbit anti-RAB7a (EPR7589), rabbit anti-CI-MPR

(ab124767), rabbit LAMP1 (ab24170), rabbit GLUT1 (ab15309), rabbit VPS35 (ab97545), rabbit VPS26 (ab181352), rabbit VPS29 (ab98929), rabbit SNX5 (EPR14358), rabbit ULK1 (EPR4885), and rabbit ATG9a (ab108338) from Abcam; mouse TOM20 (612278), mouse EEA1 (610457), mouse SNX1 (611482), and mouse SNX2 (611308) from BD Biosciences; and sheep anti-TGN46 (AHP500) and mouse anti-CI-MPR (anti-CD222, clone MEM-238) from AbD Serotec/Bio-Rad. Further antibodies used were as follows: mouse anti-GAPDH (10494-1-AP), anti-VARP (24034-1-AP), and anti-TBC1D5 (17078-1-AP) from Proteintech; SNX6 from Sigma-Aldrich (S6324); rat anti-myc (clone 9E1) and rat anti-GFP (clone 3H9) from Chromotek; and mouse anti-LAMP1 (clone H4A3) and mouse anti-LAMP2 (clone H4B4) from the DSHB. For Western blotting, all antibodies were used at a dilution of 1:1,000. For immunofluorescence, all antibodies were used at 1:100 dilution.

### Analysis of autophagic flux in VPS29 and VPS35 KO cells

To analyze autophagic flux, HeLa cells and VPS29 and VPS35 KO HeLa cells were infected with a lentivirus expressing GFP-LC3b. The cells were then seeded into 12-well plates and grown to confluence. One set of the three cell lines was left in full DMEM, whereas the other two sets were starved in EBSS for 4 h. 100 nM bafilomycin A1 (Sigma-Aldrich) was added at the beginning of the 4 h to one set of EBSS-treated cells, whereas the other set received DMSO only. The cells were then lysed in TBS, 1% Triton, and protease inhibitor cocktail, followed by Western blotting against GFP and tubulin. Autophagic flux was calculated over four independent experiments by dividing the signal intensity of the lipidated (lower) band of GFP-LC3 (signal strength was adjusted by the respective tubulin bands) in the bafilomycin-treated sample through the signal from the band of the EBSS-only sample. The flux measured in control cells was set to 100%, and the flux in the VPS KOs was calculated as a percentage of this over the four experiments.

### Statistical analysis

Results from repeat assays were analyzed for statistical significance using an unpaired *t*-test (two-sample, unequal variance) in Excel (Microsoft). Values below $P = 0.05$ were considered as statistically significant. For all experiments, we strictly compared each individual condition only with the respective control condition (e.g., parental HeLa control cells).

### RILP assay

Full-length GST-RILP protein was expressed in BL21 bacteria using pGEX-6P3 (GE Healthcare) with standard procedures and purified using GSH-Sepharose (GE Healthcare). The indicated cell lines grown to confluency in 6-cm dishes were lysed in 0.5 ml lysis buffer (Tris–HCl, pH 7.8, 50 mM NaCl, 0.5% Triton X-100, 1 mM MgCl$_2$, and Roche protease inhibitor without EDTA). 10% of the lysate was kept for the respective input analysis. GST-control beads and GST-RILP beads were then incubated with the respective lysates for 2 h at 4°C, followed by two washing steps in lysis buffer and Western blot-based detection of bound RAB7 and total RAB7 from the retained input lysates. RAB7 activity levels were calculated by dividing the bound RAB7 signal through the input RAB7 signal. The resulting

ratios were then compared across conditions by setting the HeLa control ratio to 100%. The assay was repeated at least three times for each condition shown, and the percentage activity increase/decrease was calculated over these repeat experiments. The re-addition of recombinant VPS35 was performed in the following way. GST-VPS35 was produced with the procedure described above for the GST-RILP protein. Untagged VPS35 was then cleaved from the beads with PreScission Protease (GE Healthcare) according to the manufacturer's instructions. The cleaved VPS35 was then added into the lysates from VPS35 knockout HeLa cells to a level comparable to endogenous VPS35. The RILP assay was then performed as described above.

### SILAC proteomics and RAB7 GDI assays

To screen for differential binding of GFP-RAB7 to effectors or RAB chaperones in parental HeLa cells and VPS35 KO cells, cells were cultured for 2 weeks in medium-heavy SILAC medium (Gibco, R6 and K4 amino acids from Silantes) as well as in heavy medium (R10 and K8). For a repeat experiment, the respective labels were swapped between parental HeLa cells and the VPS35 KO cells. The cells were grown to confluence in 15-cm dishes and then lysed in 1 ml of lysis buffer (Tris–HCl, pH 7.8, 50 mM NaCl, 0.5% Triton X-100, 1 mM MgCl$_2$, and Roche protease inhibitor without EDTA). GFP-RAB7 was then isolated with GFP-trap beads (Chromotek), and the IPs were combined during the final washing steps. The combined samples were then resolved on BOLT (Invitrogen) PAGE gels, digested with sequencing-grade trypsin (Promega) followed by quantification of the peptides with an Orbitrap (Thermo) and MaxQuant software. The results for GDI2 were confirmed by Western blotting for endogenous GDI2 in GFP-RAB7 precipitates (same buffer as above) from HeLa cells and VPS35 KO HeLa cells. The results were further confirmed by lentiviral expression of GFP-GDI1 in both cell lines, followed by isolation of GFP-GDI1 and Western blotting for endogenous RAB7 with endogenous RAB14 as a control.

### Quantitative Western blotting

All the Western blots in this study were acquired with a LI-COR Odyssey SA system to detect fluorescently labeled secondary antibodies. The respective band intensities were measured using the automatic background subtraction (average top and bottom setting) of the Odyssey software. All quantifications were done across at least three independent experiments as detailed in the figure legends.

### cDNA constructs

RILP, TBC1D5, RAB7, and VPS29 cDNA was cloned from HeLa cell cDNA prepared with the Superscript III kit (Invitrogen) using Kapa HiFi DNA polymerase. All site-directed mutagenesis was performed using the QuikChange method (Promega) of fully overlapping primers combined with the Kapa HiFi polymerase.

### CRISPR/Cas9

To genomically delete protein expression, the px330 plasmid (Cong *et al*, 2013) was modified with gene-specific gRNA targeting sequences according to the Zhang Lab protocol. The gRNA and CAS9-encoding px330 plasmids were co-transfected with a plasmid

encoding GFP-tagged puromycin resistance using FuGENE HD (Promega). Twenty-four hours post-transfection, cells were selected with 3 μg/ml puromycin for 24 h, followed by incubation for 3 days. Five days after transfection, 100 cells in 20 ml of DMEM or IMDM were seeded into a 96-well plate (200 μl/well) and single clones were picked when large colonies had formed. The clonal cells were screened for loss of the VPS35 protein by Western blotting. All targeting sequences were designed with the sgRNA designer of the Broad Institute (http://portals.broadinstitute.org/gpp/public/analysis-tools/sgrna-design). To disrupt VPS35 at exon 5 and exon 8, px330 plasmids targeting exon 5 and exon 8 were designed with the following gRNA sequences: GTGGTGTGCAACATCCCTTG (Ex. 5) and GAAAAGGATTCAAAGTCTGG (Ex. 8). To disrupt the VPS29 gene, three plasmids encoding VPS29 gene targeting gRNAs were mixed (GGACATCAAGTTATTCCATG, GGACATCAAGTTATTCCATG, and GGCAAACTGTTGCACCGGTG) with the puromycin resistance-encoding plasmid, and clones were isolated as described above.

For the CRISPR screen, three px330 plasmids expressing gRNAs targeting distinct genomic regions of the gene to be targeted were mixed with a plasmid encoding GFP-tagged puromycin resistance in a ratio 1:1:1:1 and transfected into the HeLa cells using FuGENE HD (Promega). One day after transfection, transfected cells were selected with 3 μg/ml puromycin (Sigma) for 24 h, followed by recovery in regular growth medium for 3 days. The cells were then split onto coverslips and analyzed for RAB7 and LAMP2 co-localization. Efficiency of the transfected CRISPR plasmids was verified by Western blotting of endogenous proteins. TBC1D5 was disrupted acutely for all of the TBC1D5-related experiments except for the RILP assay analysis of two clonal TBC1D5 knockout cells.

### Cell culture and transfection

HEK-293 and HeLa cells were grown in High Glucose DMEM (Sigma-Aldrich or Gibco) supplemented with 10% (v/v) fetal calf serum (Sigma-Aldrich) and penicillin–streptomycin (Gibco) and maintained in an incubator at 37°C and 5% $CO_2$. Cell lines were regularly tested for mycoplasma contamination. For siRNA transfection, HiPerFect (Qiagen) or Lipofectamine 2000 (Life Technologies) was used for HeLa cells. All siRNAs were made by Dharmacon (Lafayette, USA) or MWG (Munich, Germany). For DNA transfection, FuGENE 6 or FuGENE HD (Promega) was used following the manufacturers' guidelines.

### Immunoprecipitations by GFP trap

For GFP-trap IPs from HEK293 cells, 20 μg of plasmid DNA containing GFP-tagged bait was transfected into 15-cm dishes using polyethyleneimine (PEI; Polysciences) in a 1:3 ratio. Forty-eight hours post-transfection, cells were lysed in 20 mM Tris–HCl, 50 mM NaCl, 5 mM $MgCl_2$, 0.5% NP-40, and Roche EDTA-free protease inhibitor cocktail. After removing the cell debris, the lysates were incubated for 1 h with GFP-trap beads (Chromotek, Munich, Germany), followed by 2× washing in lysis buffer.

### Lentiviral expression

All lentiviruses were produced in HEK293 cells transfected with a three-plasmid system (pPAX2, pMD2G, and pLVX-puro or pXLG3).

VPS29-myc, VPS35, untagged RAB7-Q67L, and mCherry-GFP-FIS1TM were expressed with puromycin resistance from pLVX-puro (Clontech), while GFP-RAB7 and mCherry-Parkin were expressed without selection marker from pXLG3.

### Rescue experiments

For the RAB7 rescue experiments, HeLa cells were transfected with a pool of three px330 CRISPR/CAS9 plasmids expressing guide RNAs targeting three distinct regions within the RAB7a gene together with a plasmid expressing puromycin resistance. Twenty-four hours post-transfection, transfected cells were selected with 3 μg/ml puromycin for 24 h, followed by incubation for 5 days. Cells were then infected with lentiviruses expressing GFP-tagged RAB7a, RAB7a-T22N, or RAB7-Q67L and subjected to the functional assays no sooner than 3 days postinfection. For the mitophagy rescue, the GFP-RAB7-re-expressing cells were additionally infected with a lentivirus expressing mCherry-Parkin.

The VPS29 and VPS35 rescues were performed in clonal knock-out cell lines which were then infected with lentiviruses encoding VPS29-myc or VPS29-myc-L152E or untagged VPS35. For these rescues, infected cells were selected with puromycin for 2 days.

### Immunofluorescence

For immunofluorescence, cells were fixed with freshly made and ice-cold 4% PFA solution in PBS for at least 30 min, permeabilized with 0.1% saponin, and blocked with 1% BSA and 0.1% saponin in PBS for 30 min before applying indicated primary antibodies and corresponding fluorescently labeled secondary antibodies. For the methanol fixation, medium was aspirated from the cells, followed by immediate addition of pre-cooled −20°C methanol and transfer into a freezer to fix for at least 30 min at −20°C. The methanol was then removed, and cells were washed twice in PBS, followed by blocking in 1% BSA in PBS for 30 min before antibodies were applied. All images were acquired with a Leica TCS 8ST-WS confocal microscope. Images were analyzed and exported with the Volocity software package (Perkin Elmer). The Pearson's correlation between the respective channels was quantified with the co-localization tool of the Volocity software after setting of uniform thresholds across conditions. The co-localization coefficient indicated in the figures corresponds to the Mander's coefficient M1 or M2 calculated by the Volocity software after applying uniform thresholds across conditions.

### Mitochondria isolation

Parental HeLa cells and VPS35 and VPS29 KO cells were infected with a lentivirus expressing a tandem mCherry-eGFP fusion construct in frame with the C-terminal mitochondrial targeting sequence and transmembrane domain of the human FIS1 protein (amino acids 101 to end). The cells were harvested in cold sucrose buffer and disrupted by serial passage through a fine hypodermic needle, and a postnuclear supernatant was obtained by centrifuging at 800 rcf for 10 min at 4°C. The supernatants were then incubated under slow rotation with anti-GFP beads (Chromotek, Martinsried) for 1 h, followed by two washing steps with slow (1,000 rpm) pelleting of the beads. The isolated mitochondria were then lysed in

SDS–PAGE sample buffer, and the presence of RAB7 was analyzed by Western blotting. For the immunoisolation of untagged mitochondria from HeLa cells and VPS35 KO HeLa cells, a commercial mitochondria isolation kit (Miltenyi Biotec, Mitochondria Isolation Kit, human) based on antibodies against TOM22 was used according to the manufacturer's instructions. Disruption of the cells was performed with the syringe-and-needle method suggested in the instructions. The protocol was slightly modified in that the disrupted cells were centrifuged at low speed (3,000 rpm, 10 min) to remove the nuclei. Without this step, excessive amounts of nuclear DNA were found in the final purification. The success of the mitochondria isolation was confirmed by Western blotting of the isolates to detect enrichment of mitochondrial TOM20 over input levels. Markers for the ER (ERP72), lysosomes (LAMP2), and endosomes (EEA1) were not detected as enriched, whereas TOM20 (and RAB7a) was clearly enriched in the immunoisolated mitochondria fraction.

### Mitophagy assay with endogenous Parkin in SHSY-5Y cells

SHSY-5Y cells were purchased from Sigma-Aldrich and cultured in DMEM with 10% FBS and supplemented non-essential amino acids (Gibco). The cells were then infected with a lentiviral construct expressing tandem mCherry-eGFP-FIS1-TM (FIS transmembrane domain and targeting signal, AAs 152 to end) in a pLVX-puro (Clontech) backbone. Following puromycin selection of infected cells, they were transfected with siRNA against luciferase (control) and against VPS35 (Dharmacon smartpool against human VPS35) using Dharmafect-1 (Dharmacon). Efficiency of the knockdown was confirmed by Western blotting. Two days after transfection of the siRNA, cells were seeded onto fibronectin-coated coverslips and treated with the iron chelator deferiprone (dissolved to 20 mM in serum-free DMEM) at 1 mM concentration for 24 h. The cells were then fixed in 4% PFA buffered at with 50 mM HEPES, pH 7.0, for 20 min. The cells were then mounted in DAPI-containing MOWIOL mounting medium and analyzed for red-shifted punctae representing mitophagy events. This was done in two independent experiments by taking confocal images at equal strength of the GFP and the Cherry signal, followed by manual counting of red-shifted punctae. The identity of the samples had previously been obscured so that the microscope operator and manual counter were not aware of the identity of the groups.

### Fluorescence recovery after photobleaching and fluorescence loss in photobleaching

Cells stably expressing lentiviral GFP-Rab7 protein were seeded on glass-bottom culture plates (Ibidi μ) in phenol red-free DMEM (Gibco) supplemented with 5% FBS. FRAP and FLIP experiments were performed using a laser-scanning confocal microscope (LSM-I-Live Duo ZEISS LSM 510 DUO and LSM-I-UV ZEISS LSM 510 META) equipped with an environmental control system (Tokai Hit ThermoPlate System) set to 37°C and 5% $CO_2$. Five pre-bleaching images were taken and the selected area was bleached using a pulse of the 488-nm laser line at maximal intensity and 100 iterations/ROI. In FRAP experiments, photobleaching was performed using rectangular regions of interest (ROIs), and fluorescence images were acquired every second for 3 min. In FLIP experiments, the loss of fluorescence was monitored using rectangular regions of interest

(ROIs) and fluorescent images were acquired from all the regions every 3 s for 10 min. Unbleached control regions were monitored in parallel in both experiments. Signal intensity in all the selected areas was calculated using ZEN software.

### Life cell imaging

Rab7_KO HeLa cells stably expressing either lentiviral GFP-Rab7 WT, T22N, or Q67L protein growing on glass-bottom culture plates (Ibidi μ) were incubated for 15 min at 37°C and 5% of $CO_2$ with staining solution containing 250 nM MitoTracker™ Red CMXRos (Molecular Probes). After staining, cells were gently washed in 1× PBS, and 5% FBS-supplemented DMEM (Gibco) was added to the well prior to imaging with a SD-I-ABL Zeiss spinning disk inverted microscope.

**Expanded View** for this article is available online.

## Acknowledgements

We would like to thank the Life Imaging Center Freiburg (LIC), in particular Roland Nitschke and Angela Naumann, for their invaluable expertise and support in all imaging-related things. The work was funded by the Deutsche Forschungsgemeinschaft (DFG) with a grant to F.S (STE 2310/1-1).

## Author contributions

AJ-O and AK performed some of the imaging and most of the biochemical and tissue culture work. HN performed most of the molecular biology work. JDenn established the RILP assay, JDeng analyzed the GFP-RAB7 interactome by mass spectrometry, and SE provided critical expertise on RAB GTPases and helped with experimental design. FS planned the experiments, performed most of the imaging work, and wrote the manuscript.

## Conflict of interest

The authors declare that they have no conflict of interest.

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
