## [Review Process File · The EMBO Journal]

Manuscript EMBO-2017-97128

Control of RAB7 activity and localization through the Retromer-TBC1D5 complex enables RAB7-dependent mitophagy

Ana Jimenez-Orgaz, Arunas Kvainickas, Heike Nägele, Justin Denner, Stefan Eimer, Jörn Dengjel & Florian Steinberg

Corresponding author: Florian Steinberg, Albert Ludwigs Universität Freiburg

Review timeline:

Submission date:	11 April 2017
Editorial Decision:	19 May 2017
Revision received:	15 September 2017
Editorial Decision:	16 October 2017
Revision received:	18 October 2017
Accepted:	23 October 2017

Editor: Andrea Leibfried

Transaction Report:

1st Editorial Decision

19 May 2017

Thank you for submitting your manuscript for consideration by the EMBO Journal. It has now been seen by three referees whose comments are shown below.

As you will see, all three referees appreciate your work and find the mitochondrial localization of Rab7 intriguing. However, they also think that the functional role for Rab7 on mitochondria/for mitophagy remains unclear (see especially reports of referee #2 and #3).

I can offer to invite you to revise your work, but I would need really strong support from all referees on such a revised version in order to move forward here. I would thus expect much more insight into the functional role of mitochondrial-localized Rab7. Given that the outcome of addressing this is rather unclear at this stage, I took the liberty to also discuss your work with my colleague Dr. Martina Rembold at our sister journal EMBO reports. Should you not be able to provide further reaching insight, Martina will be happy to consider a revised version of your work on mitochondrial localization of Rab7 and control by retromer for publication in EMBOreports. She would expect that all technical issues are addressed (e.g. regarding imaging, co-localization, antibody specificity etc), while additional insight such as mentioned in point 8 & 9 of referee #3 would not be needed.

I hope that this slightly unusual option of further consideration of your work either at the EMBO Journal or at EMBO reports sounds like a good approach to you. Please get in touch with me to discuss the individual revision points and how you could extend your work to add further reaching insight. I am looking forward to hearing from you.

Thank you for the opportunity to consider your work for publication. I look forward to your

revision.

REFEREE REPORTS

Referee #1:

The manuscript from Jimenez-Organ et al. comprehensively describes spatial control of Rab7 activity regulated by TBC1D5 and the retromer complex. By using a newly developed Rab7 antibody, inactive Rab7 was detected on mitochondria. Upon loss of retromer subunits or TBC1D5, active Rab7 accumulates on lysosomes and cannot be localized to mitochondria anymore, which inhibits efficient mitophagy. The manuscript is well written and elucidates a novel Rab7 activity control mechanism.

Major issues:

Throughout the experiments in Figure 5, 7-8, the authors show that TBC1D5 recruitment and control of Rab7 activity are necessary for mitophagy. Their interpretation is that "VPS29 bound TBC1D5 is one of the main mechanisms to deactivate RAB7 to remove it from endo-lysosomes" (discussion section), as suggested by their rescue experiments with TBC1D5 binding deficient VPS29 mutants. However, as the authors describe themselves, TBC1D15/17 is also located to mitochondria, co-localized with Rab7 and possesses GAP activity towards Rab7. Could the authors please provide supporting data for TBC1D5 being the main GAP for Rab7 in mitophagy? As suggestion, the experiment in Fig 7A could be expanded by rescuing mitophagy deficiency in TBC1D5 KO cells with TBC1D5 wt or GAP-deficient RQ mutant variants as well as with TBC1D15. Is Rab7 localized to lysosomes upon TBC1D15/17 KO/KD (as in Fig 5A with TBC1D5 KO)?

Minor issues:

Fig 7B: Immunoblot and graph data don't fit to each other (see lanes 29KO + 29KO rescue). Are the labels mixed up?

Labeling in text: Figure 3C should be 3D and vice versa.

Labeling in text: Figure 8C should be 7C.

Labeling in text: Figure 8A is not mentioned in text.

Throughout all the figures the authors should revisit their statistical methods. Comparing more than two bars in a graph requests using one-way ANOVA.

Referee #2:

The manuscript of Jimenez-Organ et al. describes the role of the retromer complex in controlling Rab7 localization at endosomes and mitochondria. Taking advantage of a novel Rab7 antibody, the authors find endogenous Rab7 on endosomes and mitochondria. The mitochondrial localization requires the retromer and the mitochondrial GAP TBC1D5, and is lost upon inactivation or depletion of either one. Loss of the GAP does not impair retromer-mediated sorting along the endosomal system, but affects mitophagy.

This is an interesting and very carefully conducted localization study on Rab7, which reveals a number of previously unknowns. The localization of Rab7 to mitochondria is indeed surprising. The authors provide evidence that this is due to the inactivation of Rab7, and previous studies showed that Rabs accumulate on the ER (another abundant membrane) if any GEF is inactivated. Inactive Rabs may thus get deposited on the most abundant membranes. The overall principle fits with the idea that an effector (retromer) recruits the GAP to promote confinement of the Rab to a certain region and to promote its turn-over. If no recycling occurs, Rab7 is limiting for other reactions like mitophagy.

Despite my enthusiasm for the authors' impressive effort I am struggling with their overall

interpretations and implications apart from the localization of Rab7. One clear finding is the confined localization of Rab7 to mitochondria and distinct endosomal zones with the help of retromer and the GAP TBC1D5. Whether this is now a spatial control of Rab7 as a requirement for efficient mitophagy is not so clear to me. Under the selected conditions, Rab7 QL is not (efficiently) a good GAP substrate and thus not recycled from endosomes, thus not available on mitochondria (or autophagosomes?). When the GAP is lacking, the same problem occurs (as previously implied by Yamano et al., 2014). In other words, these treatments just limit the pool of available Rab7, which is of course a clear finding, yet the functional need for Rab7 on mitochondria remains unresolved.

I therefore strongly recommend that the authors describe what they have done and what they observe rather than implying from their deletions that this is essential function of retromer to control mitophagy.

Additional issues:

- 1) The authors should revise their title to reflect their data.
- 2) The Rab7 staining looks as if Rab7 is almost exclusively mitochondrial in most of their images (e.g. in Figure 3A, control). Why does it look so strongly different if the authors now look at cells with a different fixative (Figure 3B)? It would be good to control for their co-staining using anti-Rab7 with either a mitochondrial, ER or lysosomal marker. The simple reason is that I am wondering if they might pick up artifacts by their staining procedures (even though their k.o. analysis (Figure 1F, S1D) speaks against this).
- 3) The citations for the Mon1-Ccz1 complex should be corrected. The GEF activity was identified by Gerondopoulos et al, 2012 for the mammalian complex, and by Nordmann et al., 2010 for the yeast complex. The other two studies describe an involvement of both or just one protein in phagosome/endosome maturation, yet failed to assign the function.
- 4) Figure 7, text and title: Is it really that retromer is required for mitophagy? I suspect that retromer deficiency just limits the Rab7 pool (as discussed above). Thus, mitophagy is defective upon removal of retromer or TBC1D5.
- 5) Discussion: I think there is no data throughout the study to show that the Rab7 pool on mitochondria has functional importance for mitophagy. Unless they find evidence for a Rab7 acceptor on mitochondria, this pool may be just a deposit.
- 6) The Figures lack scale bars throughout, please insert.
- 7) A model would help to explain their data - also for this reviewer.

Referee #3:

Jimenez-Orgaz and colleagues describe a novel connection between Rab7 and retromer and propose retromer is a new Rab7 effector controlling Rab7 activity. They show the control of Rab7 activity by retromer results in populations of inactive Rab7 localized to the mitochondrial membrane, and an active population localized to endosomes that have retromer and lamp1 and 2 positive domains. They propose that retromer controls Rab7 activity during parkin mediated mitophagy by inactivating Rab7 and relocalizing it to mitochondria. TBC1D5 the Rab7 GAP contributes to the control of Rab7 by retromer. The authors propose the significance of the localization of Rab7 to mitochondria and control by retromer reveal a specific role for Rab7 in mitophagy. The results are very interesting and certainly important in understanding Rab7 function. It is acknowledged that very little is known about endogenous Rab7 and the authors may have found an important new tool. However, the data and the manuscript is too dense, difficult to read and many of the images are less than convincing in part because there is often so much colocalization (in Figure 2, 3 and 5). Some of the most important issues are highlighted below. In general the authors should consider reducing the number of the images shown, reducing or eliminating some data (Silac data, TBD1D15) and focusing on the main point about the control of Rab7 by retromer and mitophagy and providing more molecular insight.

General points

1. The green/red balance in many images is not done well and there is virtually no red (see S4B top panel) or cherry (see 8C) in the merge. In Figure 7 and 8 where 405 is used it is not clear what colour 405 has been false coloured to be.

2. There are no scale bars on any pictures making it impossible to judge images such as those shown in Figure 2, and zoomed in images.

Major points

1. The mitochondrial location of Rab7 should be confirmed with live cell imaging. This unexpected discovery should be now apparent in dynamic settings.
2. The Rab7 antibody specificity should be expanded by performing the Rab7 labelling in Rab7 KO cells co-labelled with TOM20, LAMP1 and/or LAMP2, and VPS35 or VPS29.
3. The authors cause confusion by mentioning methanol fixation but then don't use it as a tool or describe how it is used in methods. Do they also used formaldehyde (Fig. 4) as well as paraformaldehyde (Fig. 2).
4. What about other types of endosomes? The authors have completely ignored late endosomes/multi-vesicular bodies where Rab7 was described. Labelling with LBPA, and other markers (ALIX) should be examined to delineate the LAMP-positive domains shown in Figure 2.
5. Lamp 2 in Figure 2B looks very different from Lamp 1 in 2D. In 2B why is VPS35 apparently nuclear? Figure 2A is the same as S3A, B, C and D, 2C is the same as S3F. Why do the authors show the same cell so many times?
6. Using RLIP pulldowns the authors show that Rab7 activity is altered by loss of Vps35 and 29. But they really need to show first if retromer subunits interact with Rab7 and if this interaction affects Rab7-RLIP pulldowns.
7. In Figure 7A, VPS35 and VPS29 do not seem to have the same phenotype, although the TOM20 signal is similar. This is reflected in the loss of TOM20 in 7B with VPS29 but not VPS35.
8. To provide confidence in their model the authors should look at the role of Rab7, retromer in cells with endogenous Parkin, or independent of Parkin and a more physiological stimulus (for example loss of iron, hypoxia).
9. To provide confidence in the role of Rab7 and retromer in mitophagy the authors should look at endogenous autophagy markers (Atg9, Ulk1, WIPI2) and GABARAP instead of GFP-LC3B. A time course after induction of mitophagy would be also informative in the retromer KO.

1st Revision - authors' response

15 September 2017

Detailed response to the individual points raised by the reviewers:

Referee #1:

Major issues:

Throughout the experiments in Figure 5, 7-8, the authors show that TBC1D5 recruitment and control of Rab7 activity are necessary for mitophagy. Their interpretation is that "VPS29 bound TBC1D5 is one of the main mechanisms to deactivate RAB7 to remove it from endo-lysosomes" (discussion section), as suggested by their rescue experiments with TBC1D5 binding deficient VPS29 mutants. However, as the authors describe themselves, TBC1D5/17 is also located to mitochondria, co-localized with Rab7 and possesses GAP activity towards Rab7. Could the authors please provide supporting data for TBC1D5 being the main GAP for Rab7 in mitophagy? As suggestion, the experiment in Fig 7A could be expanded by rescuing mitophagy deficiency in TBC1D5 KO cells with TBC1D5 wt or GAP-deficient RQ mutant variants as well as with TBC1D5. Is Rab7 localized to lysosomes upon TBC1D5/17 KO/KD (as in Fig 5A with TBC1D5 KO)?

As suggested, we have performed rescue assays with re-expressed TBC1D5 (Figure 4F and Figure EV6A). Since TBC1D5 also has reported roles in autophagy that are independent of retromer, we decided to perform the rescue experiments with wildtype TBC1D5 and with the retromer binding mutant TBC1D5-L142E. As shown in Figure 4F and EV6A, mutant TBC1D5 cannot control RAB7 activity and also does not rescue the mitophagy defects. The wildtype GFP-TBC1D5 rescued both very efficiently.

Regarding TBC1D5: We have performed RILP activity assays (shown below) as well as imaging experiments (not shown) in TBC1D5 knockout cells but could not find any effect on RAB7 under steady state (as in no mitophagy) conditions. The same was observed with knockout of FIS1, which tethers TBC1D5/17 to mitochondria.

(Unpublished figure for referees not shown)

But, as reviewer 3 suggested, we have removed all data on FIS1 and TBC1D15 from the manuscript so that this data was not included into the revised version. The re-writing of the manuscript should make it much more clear that TBC1D5 and TBC1D15 work in strictly separated locations and also under different conditions. TBC1D5 constitutively switches off RAB7 on endo-lysosomes and thereby frees up pools of

available inactive RAB7. In contrast, TBC1D15 locally controls RAB7 activity strictly during mitophagy to control mitophagosome formation. We do not think that TBC1D5 is a GAP directly involved in mitophagy, but we realize that our manuscript may have been too suggestive of this. In our view, retromer bound TBC1D5 prevents lysosomal accumulation of RAB7 so that this small GTPase is available for other, non-lysosomal functions such as mitophagy and ATG9a trafficking. We specifically discuss this in the discussion to avoid giving the impression that TBC1D5 is a GAP directly involved in mitophagy. This can also be seen in the graphical synopsis that we provide with the paper.

Minor issues:

Fig 7B: Immunoblot and graph data don't fit to each other (see lanes 29KO + 29KO rescue). Are the labels mixed up?

Labeling in text: Figure 3C should be 3D and vice versa.

Labeling in text: Figure 8C should be 7C.

Labeling in text: Figure 8A is not mentioned in text.

Throughout all the figures the authors should revisit their statistical methods. Comparing more than two bars in a graph requests using one-way ANOVA.

All the above mistakes have been corrected. We are grateful for spotting this as the labels in Figure 7B were indeed mixed up, which also caused some confusion for reviewer 3 (see below). Our statistics only express significance of the indicated condition versus the control condition. We did not aim to evaluate statistically significant differences between the different KO (or rescue) conditions. We have consulted an expert on statistics who confirmed that a t-test is correct if two conditions are compared. We have amended our description of the statistical significance to make this clear.

Referee #2:

The manuscript of Jimenez-Organ et al. describes the role of the retromer complex in controlling Rab7 localization at endosomes and mitochondria. Taking advantage of a novel Rab7 antibody, the authors find endogenous Rab7 on endosomes and mitochondria. The mitochondrial localization requires the retromer and the mitochondrial GAP TBC1D5, and is lost upon inactivation or depletion of either one. Loss of the GAP does not impair retromer-mediated sorting along the endosomal system, but affects mitophagy.

This is an interesting and very carefully conducted localization study on Rab7, which reveals a number of previously unknowns. The localization of Rab7 to mitochondria is indeed surprising. The authors provide evidence that this is due to the inactivation of Rab7, and previous studies showed that Rabs accumulate on the ER (another abundant membrane) if any GEF is inactivated. Inactive Rabs may thus get deposited on the most abundant membranes. The overall principle fits with the

idea that an effector (retromer) recruits the GAP to promote confinement of the Rab to a certain region and to promote its turn-over. If no recycling occurs, Rab7 is limiting for other reactions like mitophagy.

Despite my enthusiasm for the authors' impressive effort I am struggling with their overall interpretations and implications apart from the localization of Rab7. One clear finding is the confined localization of Rab7 to mitochondria and distinct endosomal zones with the help of retromer and the GAP TBC1D5. Whether this is now a spatial control of Rab7 as a requirement for efficient mitophagy is not so clear to me. Under the selected conditions, Rab7 QL is not (efficiently) a good GAP substrate and thus not recycled from endosomes, thus not available on mitochondria (or autophagosomes?). When the GAP is lacking, the same problem occurs (as previously implied by Yamano et al., 2014). In other words, these treatments just limit the pool of available Rab7, which is of course a clear finding, yet the functional need for Rab7 on mitochondria remains unresolved.

I therefore strongly recommend that the authors describe what they have done and what they observe rather than implying from their deletions that this is essential function of retromer to control mitophagy.

We would like to thank the reviewer for this constructive and correct criticism. We have completely re-written the manuscript based on this criticism as we think that the reviewer is correct with the assessment of our data. The mitochondrial localization was so fascinating to us that we got carried away and neglected all the other endomembranes that RAB7 could readily be detected on, such as ER and Golgi membranes. We also agree that this localization to abundant endomembranes likely serves as a deposit for inactive RAB7. As the reviewer will see, we have shifted the focus of the manuscript away from the mitochondrial RAB7 to highlight the discovery of an additional function of the retromer complex in the control of RAB7 activity.

Our data demonstrate that retromer and TBC1D5 prevent lysosomal accumulation of activated RAB7, thereby freeing up pools of inactive RAB7 which then resides on various abundant endomembranes. We provide a significant amount of additional data (FRAP and FLIP assays, TBC1D5 rescue assays, loss of RAB7 from the ER, loss from the ATG9a decorated Golgi structures) to further characterize the RAB7 dysfunction upon loss of retromer. Our data clearly show that RAB7 is no longer available for non lysosomal functions as it is completely sequestered and immobilized on a swollen lysosomal compartment. It should also be clearer from our revised manuscript that retromer does not directly control RAB7 during mitophagy but instead enables mitophagy by maintaining available pools of inactive RAB7.

Additional issues:

1) The authors should revise their title to reflect their data.

Done, the title is now: "Control of RAB7 activity and localization through the Retromer-TBC1D5 complex enables RAB7 dependent mitophagy"

2) The Rab7 staining looks as if Rab7 is almost exclusively mitochondrial in most of their images (e.g. in Figure 3A, control). Why does it look so strongly different if the authors now look at cells with a different fixative (Figure 3B)? It would be good to control for their co-staining using anti-Rab7 with either a mitochondrial, ER or lysosomal marker. The simple reason is that I am wondering if they might pick up artifacts by their staining procedures (even though their k.o. analysis (Figure 1F, SID) speaks against this).

We have included new data to confirm the specificity of the RAB7a antibody and again found it to be very specific. The re-requested co-staining with TOM20 and LAMP2 are included now. We even stained a mixture of RAB7KO and Hela wildtype cells on the same coverslip, which shows that the RAB7a signal completely disappears in the KOs whereas the wildtype cells right next to the KOs show strong staining, suggesting near 100% specificity. We have also included additional data on the methanol fixation. This data should make it clear that there is no fundamental difference between PFA and methanol fixation. Being a harsher fixative, the methanol removes more of the inactive RAB7 on the ER, mitochondria and Golgi signal but tends to accentuate the vesicular RAB7. At higher laser settings, the methanol fixed RAB7 signal looks very much like that from PFA fixed cells. Because of the abundant and crowded signal on various endomembranes, we found it very hard to specifically image the vesicular pool of RAB7 in PFA fixed cells, which is why we used methanol fixation and lower laser settings.

Besides this technical aspect, we also found that the distribution of RAB7 to mitochondria, the ER and to lysosomes was highly variable and seems to depend on factors that remain to be investigated. Often, we found that the majority of RAB7 was on mitochondria and the ER, but sometimes, we found much more vesicular RAB7. This probably reflects changes in RAB7a activity state that depend on conditions that we don't yet understand. Generally, it appeared to us that nutrient depleted tissue culture medium led to a loss of RAB7 from mitochondria/ER, while fresh medium with abundant nutrients resulted in more inactive RAB7. This may reflect the level of autophagic activity or simply reflect cellular stress.

3) *The citations for the Mon1-Ccz1 complex should be corrected. The GEF activity was identified by Gerondopoulos et al., 2012 for the mammalian complex, and by Nordmann et al., 2010 for the yeast complex. The other two studies describe an involvement of both or just one protein in phagosome/endosome maturation, yet failed to assign the function.*

We have added Gerondopoulos et al., but also keep the other citations as our statement not only strictly refers to the GEF activity but also to the endosome maturation that is promoted by Mon1 and Ccz-1. Both are important studies describing a role of Mon1 or CCZ1 in endosome (or phagosome) maturation, so we still think it is appropriate to cite them in this context?

4) *Figure 7, text and title: Is it really that retromer is required for mitophagy? I suspect that retromer deficiency just limits the Rab7 pool (as discussed above). Thus, mitophagy is defective upon removal of retromer or TBC1D5.*

As stated above, this issue should be solved now and the indirect nature of the effect should be much clearer now. We have changed the title from required to "retromer enables RAB7 dependent mitophagy". We have also changed the phrasing of the figure legends. That said, we maintain that retromer is needed (or enables) for efficient mitophagy. If mitophagy is defective upon removal of retromer (as the reviewer phrases it), why is retromer not required for mitophagy? An indirect but essential role in any given process is still required for that process.

5) *Discussion: I think there is no data throughout the study to show that the Rab7 pool on mitochondria has functional importance for mitophagy. Unless they find evidence for a Rab7 acceptor on mitochondria, this pool may be just a deposit.*

We completely agree and our revised manuscript should make this much clearer.

6) *The Figures lack scale bars throughout, please insert.*

Done

7) *A model would help to explain their data - also for this reviewer.*

A model is provided in the graphical synopsis that accompanies each paper in the online version of EMBOJ.

Referee #3:

Jimenez-Orgaz and colleagues describe a novel connection between Rab7 and retromer and propose retromer is a new Rab7 effector controlling Rab7 activity. They show the control of Rab7 activity by retromer results in populations of inactive Rab7 localized to the mitochondrial membrane, and an active population localized to endosomes that have retromer and lamp1 and 2 positive domains. They propose that retromer controls Rab7 activity during parkin mediated mitophagy by inactivating Rab7 and relocating it to mitochondria. TBC1D5 the Rab7 GAP contributes to the control of Rab7 by retromer. The authors propose the significance of the localization of Rab7 to mitochondria and control by retromer reveal a specific role for Rab7 in mitophagy.

The results are very interesting and certainly important in understanding Rab7 function. It is acknowledged that very little is known about endogenous Rab7 and the authors may have found an important new tool. However, the data and the manuscript is too dense, difficult to read and many of the images are less than convincing in part because there is often so much colocalization (in Figure 2, 3 and 5). Some of the most important issues are highlighted below. In general the authors should

consider reducing the number of the images shown, reducing or eliminating some data (Silac data, TBD1D15) and focusing on the main point about the control of Rab7 by retromer and mitophagy and providing more molecular insight.

We have to really thank this reviewer as the criticism has had a major impact upon our manuscript. We are convinced that it became much better as a lot of mechanistic insight was added because of the reviewer's suggestions. We have also rewritten the manuscript completely to strictly focus on the role of retromer in the control of RAB7 activity and what this means for mitophagy. We have dropped all TBC1D15 data, as suggested, and instead provide much more mechanistic data on the RAB7 dysfunction upon loss of retromer (FRAP, FLIP, TBC1D5 rescues) and also provide mechanistic insight into the mitophagy defect using endogenous markers, which is likely (at least partially) caused by ATG9a trafficking defects. We hope that our manuscript is now more accessible.

We do not really understand the criticism of our imaging, in particular that there is too much co-localization? As we state, RAB7 completely covers the lysosomal network and the two signals (RAB7 and LAMP2) are almost identical upon loss of retromer (thresholded Pearson's correlation of 0.8!, this means near 100% overlap). This obviously results in massive co-localization irrespective of the imaging conditions. I am afraid there is nothing we can do about this.

General points

1. The green/red balance in many images is not done well and there is virtually no red (see S4B top panel) or cherry (see 8C) in the merge. In Figure 7 and 8 where 405 is used it is not clear what colour 405 has been false coloured to be.

We have evaluated the red/green balance but cannot find obvious imbalances? Overlapping red and green in our merged channels is brightly yellow with no tilt to red or green in all screened images, suggesting that red and green are in balance. Also, Figure S4B looks fine to us? There is plenty of LAMP2 (red) in S4B, also in the merged channel. We don't know what to improve here. Maybe there was an issue with our PDF? Same for Figure 8C, we clearly see mCherry-Parkin, it simply turned purple because of near 100% overlap with TOM20 in blue. We also added to the respective figure legends that the 405 channel is always shown in blue. We really think that there may have been some problem with our PDF here. We will specifically check the converted PDF of the revised version for any loss of the red channel.

2. There are no scale bars on any pictures making it impossible to judge images such as those shown in Figure 2, and zoomed in images.

We have added scale bars throughout the figures.

Major points

1. The mitochondrial location of Rab7 should be confirmed with live cell imaging. This unexpected discovery should be now apparent in dynamic settings.

We are not experts in live cell imaging and apologize for the mediocre quality of our movies. That said, the live cell imaging confirmed the localization to endomembranes such as mitochondria. We have exported single frames from the movies and also included selected movies (Movie 1-3).

2. The Rab7 antibody specificity should be expanded by performing the Rab7 labelling in Rab7 KO cells co-labelled with TOM20, LAMP1 and/or LAMP2, and VPS35 or VPS29.

We have co-labelled RAB7 in KO cells with TOM20 and with LAMP2. Figure 1B and Figure EV1F clearly prove that this antibody is 100% specific. We cannot co-label VPS35 and RAB7 in the KOs as VPS35 becomes fully cytosolic upon knockout of RAB7.

3. The authors cause confusion by mentioning methanol fixation but then don't use it as a tool or describe how it is used in methods. Do they also used formaldehyde (Fig. 4) as well as paraformaldehyde (Fig. 2).

We have included the methanol fixation in our methods now. We also provide an image demonstrating that methanol fixation and PFA fixation are not that different (Figure EV3G), it just depends on laser settings. We have not used formaldehyde, only PFA or methanol, so we have replaced formaldehyde with PFA.

4. What about other types of endosomes? The authors have completely ignored late endosomes/multi-vesicular bodies where Rab7 was described. Labelling with LBPA, and other markers (ALIX) should be examined to delineate the LAMP-positive domains shown in Figure 2.

We think that LAMP1/2 are good markers in this context as they stain both late endosomes and lysosomes. In our view, the distinction between late endosomes and lysosomes is somewhat arbitrary, as these entities undergo constant fusion and fission events. That said, we have bought an LBPA and an ALIX antibody. The former worked fine, the latter did not work at all. Co-staining with LBPA revealed that VPS35 (and also RAB7) does not really associate with LBPA positive endosomes, at least not to the extent seen with LAMP1 or LAMP2. We have included this data (Figure EV4B) but we are not sure what it means. We are somewhat uncomfortable in stating that retromer/RAB7 don't localize next to late endosomes but instead localize adjacent to lysosomes. It is clear that RAB7 covers the entire LAMP1/2 positive late endosomal/lysosomal network upon loss of retromer, so we would be happier to leave the exact nature of these vesicles open. We also have data showing that endogenous RAB7 covers LBPA positive late endosomes upon loss of retromer, but we do not think that this really provides much additional insight, so we have not included it.

5. Lamp 2 in Figure 2B looks very different from Lamp 1 in 2D. In 2B why is VPS35 apparently nuclear? Figure 2A is the same as S3A, B, C and D, 2C is the same as S3F. Why do the authors show the same cell so many times?

At least in our hands, the lysosomal network is very variable in its distribution, size of vesicles and shape. This depends on cell confluency, state of the medium and so on. Because of this, it often looks somewhat different. VPS35 is clustered around the nucleus so that in a 3D reconstruction, it appears to be nuclear. We have removed Figure 2B, though, so this issue has been solved. We have also removed all the cells that were shown twice (in different magnifications).

6. Using RLIP pulldowns the authors show that Rab7 activity is altered by loss of Vps35 and 29. But they really need to show first if retromer subunits interact with Rab7 and if this interaction affects Rab7-RLIP pulldowns.

This was very valid criticism. Retromer is a known RAB7 effector with direct binding of RAB7-GTP to VPS35, which could have an impact on the RAB7-RILP probe binding. To test this, we have produced recombinant VPS35 in bacteria and titrated this into the VPS35 KO lysate before adding the RILP beads. This had no effect on the amount of RILP bound RAB7 (Figure EV3E). We would also like to point out that our rescue experiments with mutant VPS29-L152E clearly show that it is not a loss of competition between the endogenous effector retromer (VPS35) and our RILP probe. We have added a control blot showing that mutant VPS29-L152E fully restores endogenous VPS35 levels (Figure 4B), which has no effect on the RILP bound, active RAB7. We have also added more GDI-pulldown data, FRAP and FLIP assays, all of which confirm that RAB7 behaves like a GTP locked mutant upon loss of retromer. Nevertheless, we also now state in the text that the RILP effector assay could be impacted by loss of RAB7-VPS35 binding, which we then rule out further downstream.

7. In Figure 7A, VPS35 and VPS29 do not seem to have the same phenotype, although the TOM20 signal is similar. This is reflected in the loss of TOM20 in 7B with VPS29 but not VPS35.

As Reviewer 1 pointed out correctly, we had mis-labelled the blot in Figure 7B. We apologize for that. The lack of TOM20 clearance is very similar between VPS29 and VPS35 KOs, but the distribution of the residual TOM20 is really somewhat different. We assume that this is what the reviewer refers to? We have just published that VPS29 is also an essential component of a second complex, the retriever complex (McNally et al. 2017, NCB). The loss of retriever function could cause the apparent lack of mitochondrial clustering shown in the imaging panel. That said, we have no evidence that TBC1D5 is part of the retriever complex as well, so that the VPS29 L152E mutant used throughout our study is specifically a mutant that impacts upon retromer function. We would like to point out that the TOM20 clearance phenotype in the VPS29 KO cells rescued with the L152E mutant is identical to that seen with KO of VPS35 and TBC1D5.

8. To provide confidence in their model the authors should look at the role of Rab7, retromer in cells with endogenous Parkin, or independent of Parkin and a more physiological stimulus (for example loss of iron, hypoxia).

We have used a tandem mCherry-GFP-FIS1TM mitophagy sensor in SHSY-5Y cells, which express endogenous Parkin. An evaluation of this sensor confirmed it to be an excellent indicator for mitophagy (Figure EV6B). Upon knockdown of VPS35 and Deferiprone treatment (loss of iron), we found much less red-shifting of the sensor in the VPS35 knockdown cells (Figure 6C), indicative of reduced mitophagy.

9. To provide confidence in the role of Rab7 and retromer in mitophagy the authors should look at endogenous autophagy markers (*Atg9*, *Ulk1*, *WIPI2*) and *GABARAP* instead of *GFP-LC3B*. A time course after induction of mitophagy would be also informative in the *retromer KO*.

This was excellent advice and really helped us to gain much more mechanistic insight into the mitophagy defects seen upon loss of retromer. We have performed mitophagy timecourse analyses and also used various endogenous markers (*ULK1*, *ATG9a*, *LC3b*, *ATG16L1*). This revealed that *ULK1* recruitment was normal, but *ATG9a* translocation and mitophagosome formation as evidenced by endogenous *LC3B* was defective. *GABARAPs* are not recruited to damaged mitochondria in HeLa cells (Lazarou et al., 2015, Nature), which was also the case for endogenous *gabapaps* in our hands. Interestingly, these experiments suggested that our overexpressed *GFP-LC3* greatly enhanced the speed of mitophagosome formation, as we detected complete encapsulation of the damaged mitochondria much earlier (4h) compared to the timecourse with endogenous *LC3b*. Our data quite clearly demonstrate that the hyperactivated *RAB7* is no longer able to regulate *ATG9a* trafficking, which leads to persistent TGN accumulation of *ATG9a* upon loss of retromer and induction of mitophagy. This in turn likely explains the apparent mitophagosome formation defects.

2nd Editorial Decision

16 October 2017

Thank you for submitting your manuscript for consideration by the EMBO Journal. It has now been seen by the three original referees again whose comments are enclosed. As you will see, all three referees express interest in your manuscript and are broadly in favour of publication, pending satisfactory minor revision.

I would thus like to ask you to address referee #3's remaining concern, and to provide a final version of your manuscript.

Thank you for the opportunity to consider your work for publication. I look forward to your revision.

REFeree REPORTS

Referee #1:

The authors have added a tremendous amount of experimental data and rigorously revised their manuscript in response to the reviewers comments. All my concerns have been adequately addressed and I recommend to accept this manuscript for publication. Good job!

Referee #2:

I went through the revised manuscript. The authors nicely answered my questions and rewrote the manuscript in a much more consistent manner. Their data is of high quality and suggest a major role of Retromer as a regulator of Rab7 localization in mammalian cells. I thus have no further questions and recommend the study for publication.

Referee #3:

The authors have improved the manuscript in many different areas. While it is still very dense and somewhat hard to read, it is better and the data will be of interest to many researchers. I still find the images to be in some cases overly bright or too highly contrasted (eg 6D, 7B, 8A (TOM20)), and would have preferred larger areas of the cell (eg 7B and 8D) but the data is still valid.

Only two major points: 1) the second half of the abstract needs rewriting, in particular lines 6-9 make no sense. The use of "deposits" is unfortunate because indeed in my opinion some of the over exposed labelling on the cell organelles do resemble "deposits" of proteins. 2) "mitophagosomes"

imply that these are not the same compartment as autophagosomes. The authors should consider the fact they are studying mitophagy which requires the formation of autophagosomes selectively targeting mitochondria.

2nd Revision - authors' response

18 October 2017

Thank you very much for the constructive handling of our manuscript. We are delighted that our manuscript will be published in the EMBOJ. We have now submitted what we hope will be the final version of our manuscript. As suggested by Reviewer 3, we have changed the wording of the abstract.

3rd Editorial Decision

23 October 2017

Thank you for submitting the final revision of your manuscript to us. I appreciate the introduced changes, and I am happy to accept your work for publication in the EMBO Journal. Congratulations!

Corresponding Author Name: Florian Steinberg

Journal Submitted to: EMBOJ

Manuscript Number: EMBOJ-2017-97128